# Hyperinflammatory repolarisation of ovarian cancer patient macrophages by anti-tumour IgE antibody, MOv18, restricts an immunosuppressive macrophage:Treg cell interaction

Ovarian cancer is the most lethal gynaecological cancer and treatment options remain limited. In a recent first-in-class Phase I trial, the monoclonal IgE antibody MOv18, specific for the tumour-associated antigen Folate Receptor-α, was well-tolerated and preliminary anti-tumoural activity observed. Pre-clinical studies identified macrophages as mediators of tumour restriction and pro-inflammatory activation by IgE. However, the mechanisms of IgE-mediated modulation of macrophages and downstream tumour immunity in human cancer remain unclear. Here we study macrophages from patients with epithelial ovarian cancers naive to IgE therapy. High-dimensional flow cytometry and RNA-seq demonstrate immunosuppressive, FcεR-expressing macrophage phenotypes. Ex vivo co-cultures and RNA-seq interaction analyses reveal immunosuppressive associations between patient-derived macrophages and regulatory T (Treg) cells. MOv18 IgE-engaged patient-derived macrophages undergo pro-inflammatory repolarisation ex vivo and display induction of a hyperinflammatory, T cell-stimulatory subset. IgE reverses macrophage-promoted Treg cell induction to increase CD8+ T cell expansion, a signature associated with improved patient prognosis. On-treatment tumours from the MOv18 IgE Phase I trial show evidence of this IgE-driven immune signature, with increased CD68+ and CD3+ cell infiltration. We demonstrate that IgE induces hyperinflammatory repolarised states of patient-derived macrophages to inhibit Treg cell immunosuppression. These processes may collectively promote immune activation in ovarian cancer patients receiving IgE therapy.

Ovarian cancer has the highest mortality rate of all gynaecological cancers[1]. This poor prognosis is promoted by processes such as early peritoneal metastasis and resistance to therapies and anti-tumoural immunity[2]. Tumour-associated macrophages (TAMs) are highly abundant in the ovarian tumour microenvironment (TME) and are reported to contribute extensively to these pathological processes[3–11]. Although TAM tumour infiltration does not hold prognostic significance[12], infiltration by TAMs polarised to alternatively-activated states inversely correlates with patient survival[13,14]. Consequently, targeted therapies to engage TAMs and repolarise them to

✉ e-mail: sophia.karagiannis@kcl.ac.uk

pro-inflammatory, anti-tumoural states represent a strategy of interest for ovarian cancer.

Tumour antigen-specific monoclonal antibodies (mAbs) offer such a targeted approach by binding to TAM Fc receptors via their Fc regions and simultaneously to tumour cells. This can induce TAM activation and tumour cell killing[15]. Hitherto, tumour antigen-specific mAbs have almost exclusively utilised the IgG isotype; however, poor clinical results in ovarian cancer have resulted in none receiving UK regulatory approval[16,17]. Consequently, novel mAb strategies are required.

The use of an IgE antibody isotype offers the possibility of engaging a previously untapped immune axis in oncology, in the emerging field of AllergoOncology[18]. IgE antibodies possess unique attributes that could be of therapeutic utility in ovarian cancer. IgE has a higher affinity for, as well as a slower dissociation rate from, its high-affinity receptor FcεRI, compared to IgG:Fcγ receptor (FcγR) interactions[19,20]. This should facilitate quicker trafficking to and retention at the TME, with a reported tissue half-life for IgE of ~14 days compared to ~3 days for IgG[21,22]. IgE can trigger rapid, pro-inflammatory immune reactions, which require only partial occupancy of antigen-bound antibody to Fc receptors[23]. These reactions frequently occur in protective immune responses, such as parasite clearance, within alternatively-activated immune environments[24], which may bear similarity to the ovarian TME. Furthermore, IgE has no known inhibitory Fc receptor, compared to CD32b for IgG, which is frequently expressed by TAMs and likely contributes to therapeutic resistance for IgG mAbs[25]. Accordingly, there is growing evidence for IgE-mediated anti-tumoural surveillance, including negative correlations between serum IgE levels and gynaecological cancer risk[26].

MOv18 IgE, which targets the tumour-associated antigen Folate Receptor-α (FRα), is the first-in-class to enter clinical testing[27]. In pre-clinical studies, MOv18 IgE displayed anti-tumoural efficacy through a monocyte/macrophage-mediated mechanism[28,29]. In a rat syngeneic lung metastases model, superior metastasis restriction was observed for rat MOv18 IgE compared to its IgG2b counterpart and was associated with increased intra-tumoural TAM migration[29]. In the first-in-human Phase I trial (NCT02546921), MOv18 IgE administration was well tolerated and showed preliminary evidence of anti-tumoural activity at a low dose[27].

The monocyte/macrophage-mediated mechanism of MOv18 IgE is proposed to be two-armed, comprising tumour cell killing and pro-inflammatory activation[30]. With respect to the second arm, MOv18 IgE triggered a pro-inflammatory TNF/CCL2-mediated positive feedback loop of monocyte/macrophage tumour recruitment and activation[29]. Additionally, IgE induced a pro-inflammatory repolarisation signature in alternatively-activated M2 macrophages[31], associated with improved ovarian cancer patient prognosis[32].

Macrophage modulation by IgE within ovarian cancer, however, remains unclear. Here, we study IgE stimulation of ovarian cancer patient-derived macrophages and the downstream impact on T cells as key components within the TME immune compartment. Specifically, we characterise patient-derived macrophages by high-dimensional flow cytometry analysis of patient ascites-conditioned macrophages and TAMs isolated from patient ascites, integrated with public RNA-seq datasets of treatment-naïve patient tumours. We evaluate how these macrophages compare to established in vitro-derived human macrophage subsets, express FcεRs, and bind to MOv18 IgE ex vivo. We assess their stimulation by IgE ex vivo, through characterisation of tumour cell killing and phenotypic repolarisation at a subset level. In co-cultures and RNA-seq cell:cell interaction analyses, we evaluate the interactions between patient-derived macrophages and regulatory T (Treg) cells and how their modulation by IgE affects CD8+ T cell expansion. Finally, we characterise whether this IgE-driven immune signature is recapitulated in vivo following MOv18 IgE treatment in a syngeneic rat model of lung metastasis and in pre-treatment and on-treatment tumour biopsies from patients in the Phase I trial of MOv18 IgE.

## Results

### IgE Fc engagement drives a net pro-inflammatory shift in a panel of in vitro-derived human macrophages

Since a macrophage-mediated mechanism was implicated in tumour growth restriction by IgE in rodent models[28,29], we first investigated whether IgE can engage and activate human macrophages across different polarisation states.

To model the spectrum of human macrophage polarisation, we utilised a panel of in vitro-derived macrophage subsets with known phenotypes and functions, which expands upon the binary classically-activated (M1) / alternatively-activated (M2) polarity model. This panel comprised: IFNγ/LPS-polarised pro-inflammatory M1; IL-4-polarised Th2-skewed M2a; IgG1-cross-linked immunoregulatory M2b; IL-10-polarised immunosuppressive M2c and adenosine-polarised angio-genic M2d (Supplementary Fig. 1a–c). Each subset displayed the expected surface marker and secreted factor profiles ($n = 3$–$9$) (Fig. 1a and Supplementary Fig. 1c) and exhibited distinct phenotypes ($n = 2$) (Fig. 1b).

IgE receptors, high-affinity FcεRI ($n = 9$) and low-affinity CD23 ($n = 7$), were expressed differentially across polarisation states: IL-4-polarised M2a exhibited the highest expression; IL-10 M2c polarisation was associated with high expression, whilst IFNγ/LPS-polarised M1 displayed negligible expression (Fig. 1c). Accordingly, all subsets apart from M1 displayed binding to MOv18 IgE ($n = 3$) (Fig. 1d).

Macrophage activation by IgE was investigated using polyclonal anti-human IgE antibodies, which cross-link FcεR-bound IgE by mimicking immune complex formation (Supplementary Fig. 1a). This included cross-linking of endogenous IgE only (anti-IgE), retained on macrophage FcεRs following ex vivo derivation[33,34], and of both endogenous IgE and exogenously applied MOv18 IgE (MOv18 IgE + anti-IgE) (Fig. 1e). Consistent with FcεR expression, FcεR:IgE cross-linking triggered pro-inflammatory activation in alternatively-activated subsets M2a and M2c, while M1 retained their pro-inflammatory phe-notype. M2b and M2d, exhibiting lower FcεR levels, were unresponsive under these stimulation conditions. IgE-mediated repolarisation of M2a and M2c comprised a downregulation of scavenger receptors CD163, CD204 and MerTK ($n = 9$), which promote anti-inflammatory macrophage phenotypes, and an upregulation of T cell co-stimulatory molecules, including CD40, CD80 and CD86 ($n = 5$). Accordingly, FcεRI was downregulated only in these IgE-responsive subsets ($n = 6$). tSNE dimensionality reduction confirmed global phenotype shifts for M2a and M2c only ($n = 2$) (Fig. 1f). These cell surface marker profile mod-ulations, combined with the broad secretome activation detected in cell culture supernatants ($n = 8$) (Luminex) (Supplementary Fig. 1d), demonstrated comprehensive phenotypic and functional shifts for M2a and M2c following FcεR:IgE cross-linking ($n = 3$–$9$) (Fig. 1g). This shift was equivalent between cross-linking of endogenous IgE and of both endogenous IgE and MOv18 IgE, as previously described[31,32], indicating a maximal repolarisation effect via cross-linking only a fraction of FcεRs.

Overall, IgE engages and activates alternatively-activated macro-phage subsets to drive a net pro-inflammatory macrophage phenotype shift.

### Ovarian TME is enriched for alternatively-activated macrophage phenotypes, which associate with poor patient survival, *IL10* signaling and *FCER1A*/FcεRI expression

In view of the pro-inflammatory repolarisation of alternatively-activated macrophages by IgE, we next characterised the phenotype of patient-derived macrophages in ovarian cancer and their potential for IgE activation, firstly via transcriptomic analysis.

The CIBERSORTx deconvolution algorithm was used to estimate immune cell abundance in treatment-naïve primary ovarian tumours (TCGA-OV; $n = 378$; bulk RNA-seq) (Fig. 2a). Hierarchical clustering demonstrated that tumours did not separate into immune 'hot' and

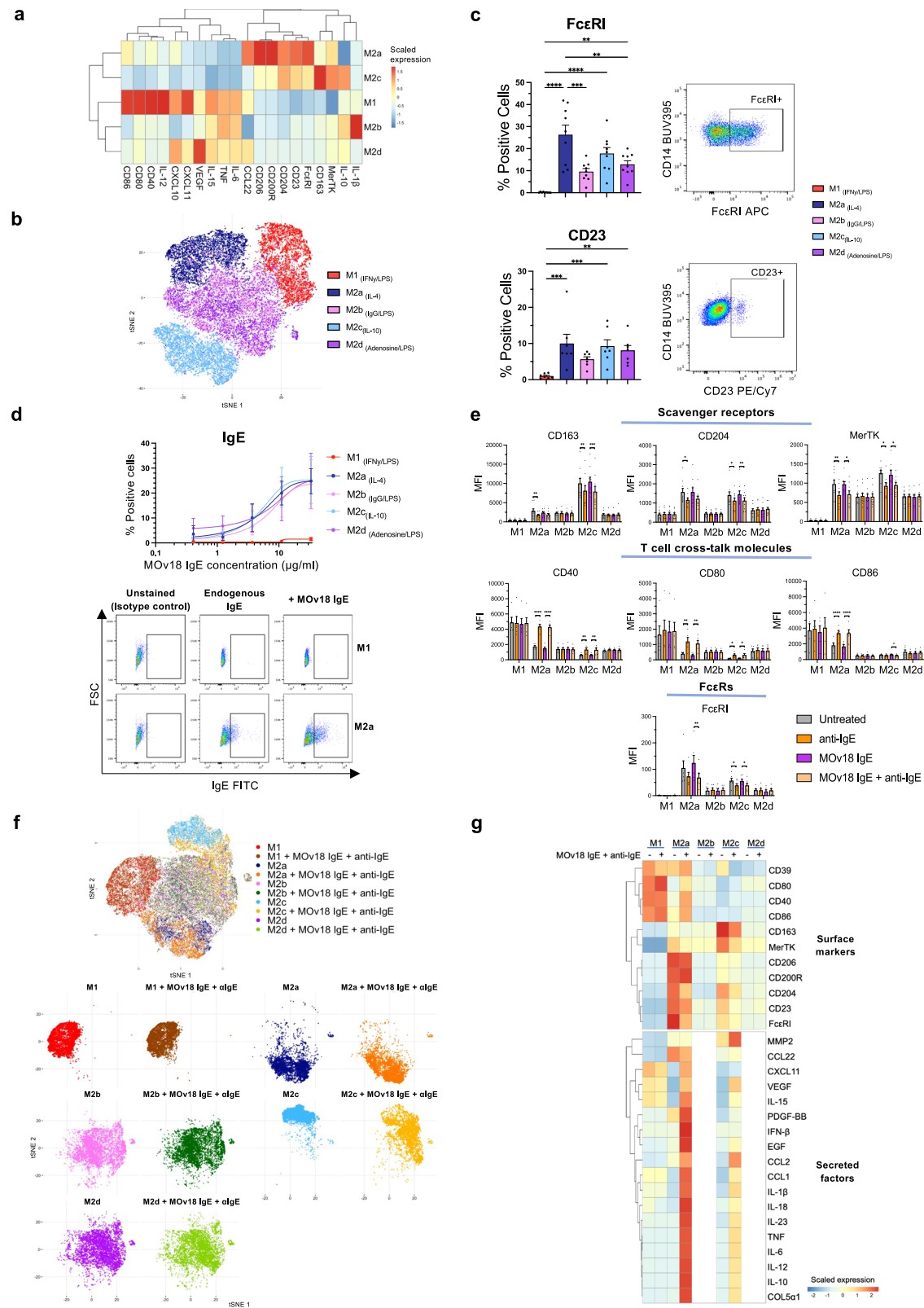

'cold'. Macrophages were the most abundant cell type (42.5 ± 0.7 (SEM)%). Additionally, CD8+ T cells (7.9 ± 0.3 (SEM)%) and Natural Killer (NK) cells (18.2 ± 0.4 (SEM)%) were present, indicating the potential for harnessing anti-tumuoral immune responses in patients. Patient stratification by tumour immune cell abundance (Immune Score) or by tumour expression of the pan-macrophage gene *CD68* lacked prognostic significance (Fig. 2b). However, expression of the

alternatively-activated macrophage gene *CD163* was inversely correlated with survival (Fig. 2b). Analysing tumours by the binary classically-activated M1 and alternatively-activated M2 macrophage polarity (CIBERSORTx), revealed higher infiltration by M2 macrophages (Supplementary Fig. 2a). Prognostically, M2^high tumours associated with poorer patient survival, whilst the inverse association was found for M1^high tumours (Supplementary Fig. 2b).

**Fig. 1 | In vitro-derived human macrophage subsets express FcεRs, bind MOv18 IgE and exhibit a net pro-inflammatory phenotypic shift following IgE stimulation.** Evaluation of in vitro-derived human macrophage subsets FcεR expression, MOv18 IgE binding and phenotypes following FcεR:IgE cross-linking by polyclonal anti-IgE antibodies. **a** Heatmap displaying the scaled expression of surface markers (MFI) (flow cytometry) ($n = 9$ CD86, CD80, CD40, CD206, CD204, CD163, MerTK; $n = 8$ FcεRI; $n = 7$ CD23; $n = 3$ CD200R) and secreted factors (pg/ml) in cell culture supernatants (Luminex) ($n = 5$). **b** tSNE plot visualising the in vitro-derived macrophage subsets based on surface marker expression (flow cytometry) ($n = 2$). **c** Comparison of FcεRI ($n = 9$) and CD23 ($n = 7$) expression, with representative flow cytometry plots for M2a macrophages. **d** Assessment of MOv18 IgE binding ($n = 3$), with representative flow cytometry plots for M1 and M2a macrophages. **e** Comparison of surface marker expression following FcεR:IgE cross-linking (anti-IgE: cross-linking endogenous IgE only; anti-IgE + MOv18 IgE: cross-linking endogenous IgE and exogenously applied MOv18 IgE) (flow cytometry) ($n = 9$ CD163,

CD204, MerTK; $n = 6$ FcεRI; $n = 5$ CD40, CD80, CD86). **f** Composite and faceted tSNE plots visualising the in vitro-derived macrophage subsets based on surface marker expression following FcεR:IgE cross-linking (flow cytometry) ($n = 2$). **g** Following detection of IgE-mediated repolarisation of M2a and M2c, the effect of FcεR:IgE cross-linking on the secretome of M2a and M2c, as well as M1, was assessed by Luminex measurement of secreted factors in macrophage culture supernatants. Heatmap displaying the scaled expression of surface markers (MFI) (flow cytometry) ($n = 9$ CD86, CD80, CD40, CD206, CD204, CD163, MerTK; $n = 8$ FcεRI; $n = 7$ CD23; $n = 5$ CD39; $n = 3$ CD200R) and secreted factors (pg/ml) in cell culture supernatants (Luminex) ($n = 8$) following FcεR:IgE cross-linking. Data shown as mean ± SEM. Statistical significance was calculated using a repeated measures 1-way ANOVA with Tukey's post hoc test (**c**, **e**); *$P_{adj} <0.05$, **$P_{adj} <0.01$, ***$P_{adj} <0.001$ and ****$P_{adj} <0.0001$. Source data and exact P/$P_{adj}$ values are provided as a Source Data file.

Gene set enrichment analysis (GSEA) was used to delineate which pathways might underpin this potentially deleterious alternative activation of macrophages. *IL10* signaling was found to be the cytokine pathway of greatest enrichment in *CD163*high tumours (Reactome) (Fig. 2c).

Consistent with FcεRI expression and IgE-mediated pro-inflammatory repolarisation of CD163high in vitro-derived IL-10-polarised M2c (Fig. 1), we identified enrichment for the high-affinity IgE receptor gene, *FCER1A*, in *CD163*high tumours (Fig. 2d). Moreover, single-cell RNA-seq (scRNA-seq) analysis (Zhang et al.[35], GSE165897, $n = 6$) of treatment-naïve ovarian cancer metastatic peritoneal tumours ($n = 7124$ immune cells) identified monocytes and macrophages to exhibit highest *FCER1A* expression, with dendritic cell (DC) types also showing low expression (Fig. 2e). We identified 1 monocyte and 5 distinct macrophage subsets in tumours ($n = 1374$) (Supplementary Fig. 2c) and confirmed the presence of *FCER1A* + TAMs consistently across patients (Fig. 2f). *FCER1A* expression at the transcriptomic level was recapitulated by flow cytometric analysis of ovarian cancer patient ascites. Ascites TAMs exhibited consistent expression of FcεRI ($n = 14$) (Fig. 2g, top) as well as CD23 ($n = 14$) (Supplementary Fig. 2d). Furthermore, consistent with scRNA-seq analyses (Fig. 2e), TAMs were highly abundant in ascites ($19.7 \pm 4.9$ (SEM)% of CD45+ cells) ($n = 11$), representing the most frequent FcεRI-expressing immune cell type, whilst low levels of DCs ($n = 8$) and absent mast cells ($n = 4$) were detected (Fig. 2g, bottom, Supplementary Fig. 2e).

In summary, ovarian tumors are enriched for alternatively-activated macrophage phenotypes, which associate with poor patient survival and *IL10* signaling. However, macrophage FcεRI expression may provide an opportunity to influence their phenotypes by IgE stimulation.

## Ovarian cancer patient ascites-conditioned macrophages (MAsc) and ascites TAMs exhibit strong phenotypic similarity to IL-10 polarised macrophages

We next further characterised the polarisation effect of the ovarian TME on macrophages via ex vivo flow cytometry analysis.

Monocytes from healthy volunteers were differentiated to macrophages in the presence of ascites from 19 ovarian cancer patients (Fig. 3a). Compared to unpolarised M0 macrophages, ascites-conditioning (MAsc) triggered an upregulation of anti-inflammatory scavenger receptors CD163 and MerTK (Fig. 3b). Additionally, an immunosuppressive shift in T cell cross-talk molecules was observed, comprising downregulation of antigen presentation markers HLA-DR and CD80 and upregulation of checkpoint ligand PD-L1 (Fig. 3b). Comparison to in vitro-derived macrophage subsets via hierarchical clustering and UMAP dimensionality reduction, revealed that MAsc exhibited strongest phenotypic similarity to IL-10-polarised immunosuppressive M2c ($n = 19$ MAsc, $n = 6$ in vitro-derived macrophages) (Fig. 3c, d). This similarity was driven principally by expression of the

CD163, MerTKhigh CD40, CD80low signature (Fig. 3d), which defines M2c polarisation (Fig. 1a, and Supplementary Fig. 1c).

We subsequently characterised TAMs isolated from patient ascites ($n = 10$) (Supplementary Fig. 3a). Hierarchical clustering of ascites TAMs, MAsc and the full in vitro-derived macrophage panel, demonstrated that TAMs, like MAsc, exhibited strong similarity to IL-10-polarised M2c, with TAMs also exhibiting an M2c-associated immunosuppressive CD163high, CD40, CD80low signature (Fig. 3e). MAsc and TAMs also showed low expression of additional T cell co-stimulatory molecule CD86 (Fig. 3e).

In concordance with our identification of a prevailing immunosuppressive macrophage polarisation in the ovarian TME, scRNA-seq pseudotime analysis of TAMs from metastases of the same intraperitoneal space as ascites projected an immunosuppressive TAM differentiation trajectory (Fig. 3f, g). Following monocyte to macrophage differentiation, mature TAMs could first transition to a subset enriched for classically-activated, immunostimulatory processes (MHC II TAMs), as determined by gene over-representation analysis (Reactome, Gene Ontology Biological Processes (GOBP)). This included pathways for MHC Class II Antigen Presentation, *IFNG* Signaling and Positive Regulation of Lymphocyte Activation. However, downstream, TAM subsets were associated with an immunosuppressive transition, similar to MAsc and ascites TAMs. This comprised a downregulation and eventual loss of the immunostimulatory functional enrichment and a concurrent upregulation of *IL10* signaling, as MHC II TAMs transitioned to Tissue Remodelling TAMs and finally to Hypoxic TAMs. Interestingly *FCER1A*+ TAMs belonged to a distinct trajectory that did not involve further downstream differentiation.

In summary, consistently across primary and metastatic tumours and ascites, ovarian cancer macrophages exhibit immunosuppressive phenotypes in accordance with IL-10 polarisation.

## MOv18 IgE-engaged patient-derived macrophages kill ovarian tumour cells and undergo a pro-inflammatory repolarisation

In view of the similarity of patient-derived macrophages to immunosuppressive IL-10-polarised M2c, we investigated whether MAsc and ascites TAMs might undergo a similar IgE-mediated pro-inflammatory repolarisation as M2c ex vivo (Fig. 4a).

Both MAsc ($n = 4$) and ascites TAMs ($n = 2$) exhibited binding to MOv18 IgE (Fig. 4b). MAsc showed a pro-inflammatory repolarisation following FcεR:IgE cross-linking ($n = 19$) (Fig. 4c). This reversed their phenotypic similarity to M2c which was induced by ascites-conditioning, through IgE-mediated CD163 and MerTK downregulation and CD40 and CD80 upregulation. FcεR:IgE cross-linking of MAsc ($n = 19$) also upregulated PD-L1 (Fig. 4c) and triggered a broad secretome activation, similar to that observed for IgE-activated M2c ($n = 19$) (Fig. 4d and Supplementary Fig. 3b), whilst HLA-DR expression was unaffected (Supplementary Fig. 3c). tSNE and MDS dimensionality reduction demonstrated a global IgE-mediated pro-inflammatory repolarisation of MAsc across donors (Fig. 4e). This IgE-mediated

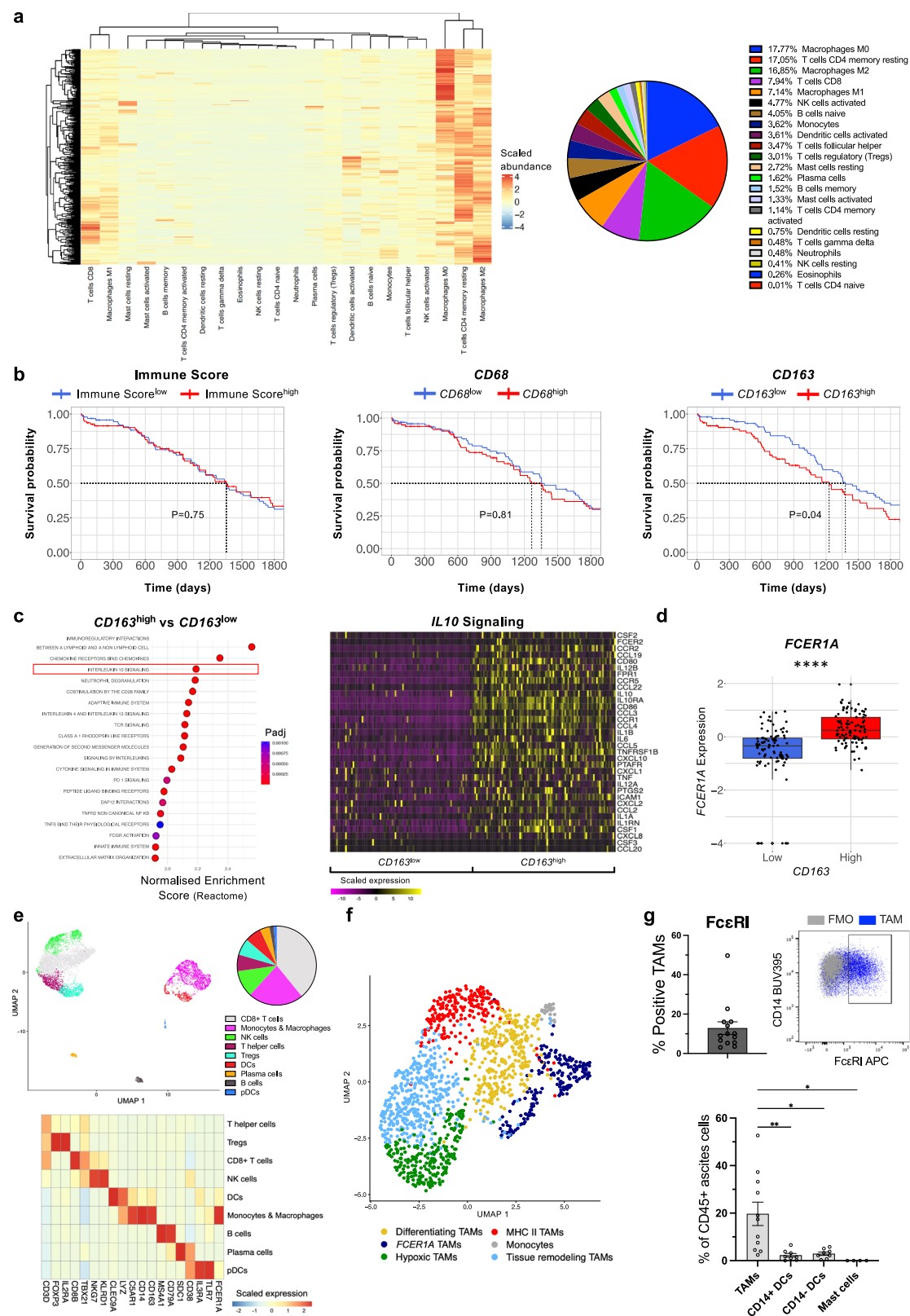

repolarisation signature, of downregulated scavenger receptors and upregulated T cell cross-talk molecules, was replicated by target specific cross-linking of MAsc-bound MOv18 IgE, in MAsc co-cultures with FRα + IGROV1 ovarian cancer cells (n = 6) (Fig. 4f).

Ascites-isolated TAMs also exhibited a comparable repolarisation signature, following FcεR:IgE cross-linking, as seen with M2c and MAsc (n = 13) (Fig. 4g). This comprised an upregulation of T cell cross-talk

molecules and a downregulation of specific scavenger receptors, including a reversal of their M2c-associated CD163, MerTK[high] CD40, CD80[low] immunosuppressive signature. TAM HLA-DR expression trended towards upregulation following FcεR:IgE cross-linking, but did not reach statistical significance (Supplementary Fig. 3d). We found IgE to be consistently detectable in patient serum (n = 15) and ascites (n = 20), with lower levels in ascites (Supplementary Fig. 4a).

**Fig. 2 | The ovarian tumour microenvironment (TME) is enriched for alternatively-activated macrophage phenotypes, which associate with poor patient survival, *IL10* signaling and *FCER1A*/FcεRI expression.** Bulk (TCGA-OV; *n* = 378) and single-cell (GSE165897; *n* = 6) RNA-seq and flow cytometric evaluations of the TME of ovarian cancer patients. **a** Heatmap displaying the scaled abundance of immune cells in TCGA-OV primary tumours as determined by the CIBERSORTx deconvolution package and a pie chart representing mean immune cell abundance across patients. **b** Kaplan−Meier plots (Survival and Survminer packages) stratifying TCGA-OV patients by high and low levels (quartiles) of tumour immune cell abundance (Immune Score; ConsensusTME deconvolution package) and tumour *CD68* and *CD163* expression. **c** Left: Following patient stratification by *CD163* tumour expression (*n* = 190), differentially expressed genes (DEGs) were determined using the DESeq2 package and enrichment of genes sets evaluated within the human Reactome database (v2023.1) by the fgsea package. Graph displaying the top 20 upregulated Reactome pathways ordered by Normalised Enrichment Score, with Interleukin 10 Signaling highlighted. Right: Heatmap displaying the scaled expression (TPM: transcripts per million) of genes in the Interleukin 10 Signaling pathway (*n* = 34 genes). **d** Boxplot comparison of *FCER1A* expression

($\log_{10}$(TPM + 0.001)) between *CD163*^high and *CD163*^low tumours (*n* = 190) (*P* < 0.0001). **e** Top: UMAP visualising the annotated immune cell types in GSE165897 (metastatic peritoneal tumours; *n* = 7142 cells) following unsupervised clustering using the Seurat package and a pie chart displaying mean cell type abundance across patients. Bottom: Heatmap displaying the scaled expression (TPM) of lineage marker genes used for cell type annotation and *FCER1A*. **f** UMAP visualising the annotated monocyte and macrophage subsets in GSE165897 (*n* = 1374 cells) following unsupervised clustering. **g** Flow cytometry evaluation of TAM FcεRI expression (*n* = 14) (top) and TAM (*n* = 11), dendritic cell (DC) (*n* = 8) and mast cell (*n* = 4) frequencies in patient ascites (bottom). Data shown as median (centre line), interquartile range (IQR) (box) and range within 1.5 x IQR (whiskers) (**d**) and mean ± SEM (**g**). Statistical significance was calculated using a Flemington-Harrington weighted log rank test (**b**), Wald test with Benjamini−Hochberg correctionh (**c - DEGs**), permutation testing (**c - gene sets**), a two-tailed Wilcoxon signed rank test (**d**) and a mixed effects analysis with Tukey's post hoc test (**g**); *P/Padj <0.05, **P/Padj <0.01 and ****P/Padj <0.0001. Source data and exact P/Padj values are provided as a Source Data file.

This consistent presence of endogenous IgE antibodies is accordant with the activation of MAsc and ascites TAMs when cross-linking endogenously-bound IgE only.

Additionally, antigen-specific cross-linking by MOv18 IgE triggered anti-tumoural and pro-inflammatory activation of TAMs. scRNA-seq evaluation of metastatic peritoneal tumours (Supplementary Fig. 4b) and flow cytometric evaluation of ascites (*n* = 2) (Supplementary Fig. 4c) demonstrated no *FOLR1*/FRα expression on TAMs, whilst tumour cells expressed high levels at an RNA and protein level, respectively. Accordingly, TAMs mediated antibody-dependent cellular cytotoxic (ADCC) killing of IGROV1 cancer cells by MOv18 IgE, which was associated with CD163 downregulation on TAMs (*n* = 5) (Fig. 4h).

Furthermore, in view of the presence of FcεRI + DCs at a low level in the ovarian TME (Fig. 2e, g bottom), we also assessed their capacity for MOv18 IgE-mediated activation ex vivo. MOv18 IgE bound to DCs (*n* = 4) (Supplementary Fig. 5a) and enhanced their phagocytosis of FRα-coated beads compared to an IgE isotype control (*n* = 5) (Supplementary Fig. 5b). Additionally, very low levels of basophils were identified in patient ascites (*n* = 5) (Supplementary Fig. 5c), but when stimulated ex vivo, they were capable of IgE-mediated activation, as determined by upregulation of the activation marker CD63 (*n* = 5) (Supplementary Fig. 5d).

### IgE-mediated repolarisation shifts patient-derived macrophage subsets away from immunosuppression towards an IgE-induced hyperinflammatory subset

Subset level characterisation of this IgE-mediated repolarisation of patient-derived macrophages was then conducted. Firstly, at baseline, MAsc (*n* = 19) displayed an enrichment of 2 immunosuppressive subsets, *e* and *k*, following ascites conditioning (Fig. 5a−d). These subsets expressed high levels of CD163 and MerTK (subset *e*), and expression profiles of T cell cross-talk molecules skewed towards immunosuppression: enriched PD-L1; absent CD80 and low (subset *e*) or absent (subset *k*) CD40. Moreover, the FcεRI+ MAsc subset, *a*, had a similarly immunosuppressive phenotype, lacking CD40, CD80 and HLA-DR and exhibiting low CD86 expression (Fig. 5a−d).

FcεR:IgE cross-linking drove a global shift in MAsc subsets towards pro-inflammatory activation, downregulating ascites-induced immunosuppressive subsets, *e* and *k*, and simultaneously enriching 3 pro-inflammatory subsets, *c*, *h* and *j* (Fig. 5a−d). Subsets *c* and *j*, present at negligible levels in both M0 and MAsc, were induced by IgE stimulation. Subset *j* exhibited a hyperinflammatory phenotype, featuring high expression of all T cell activation molecules assessed, HLA-DR, CD40, CD80, CD86, as well as high PD-L1, FcγR CD16 and absent MerTK.

Similar subset-level findings were observed for IgE-stimulated ascites TAMs (*n* = 7). Highest enrichment of FcεRI and CD23 was

identified on an immunosuppressive subset, *13*, which exhibited high scavenger receptor expression and absent CD80 (Fig. 5e, f). FcεR:IgE cross-linking upregulated a highly similar hyperinflammatory subset as that observed for MAsc, subset *7*, which also displayed high expression of all T cell activation molecules assessed, HLA-DR, CD40, CD80, CD86 (Fig. 5e−g).

These findings indicate that IgE stimulation can drive a repolarisation of patient-derived macrophage subsets away from immunosuppression, towards an IgE-induced hyperinflammatory subset enriched for T cell stimulation.

### IgE-mediated hyperinflammatory repolarisation of patient-derived macrophages reverses an immunosuppressive macrophage:Treg cell interaction to promote CD8+ T cell expansion

In view of the immunosuppressive phenotypes of patient-derived macrophages and their association with poor prognosis, we next investigated whether macrophages exert pro-tumoural effects on the wider TME immune compartment and the scope for IgE-mediated modulation of these effects.

Cell:cell interaction analysis of immune cells from metastatic tumours (scRNA-seq) found TAMs to exhibit the highest interaction frequency (sent and received) (Fig. 6a). Treg cells were the lymphocyte in receipt of the most TAM-derived interactions. Accordingly, in primary tumours (bulk RNA-seq), expression of the alternatively-activated macrophage gene *CD163* positively correlated with Treg master transcription factor *FOXP3* (Fig. 6b). Consistent with this, *CD163*^high primary tumours were enriched for the Positive Regulation of Treg Differentiation pathway (GSEA, Reactome) (Fig. 6c) and Treg cell lineage genes *FOXP3* and *CTLA4* (Supplementary Fig. 6a). Treg cells were present in primary (bulk RNA-seq, CIBERSORTx) and metastatic (scRNA-seq) tumours, representing 9.4 ± 0.4 (SEM)% and 17.2 ± 6.2 (SEM)% of T cells, respectively (Fig. 2a, e).

Based on infiltration of and associations between macrophages and Treg cells in tumours, we subsequently investigated a potential immunosuppressive macrophage:Treg cell interaction and characterised its scope for IgE-mediated modulation in co-culture assays (Fig. 6d and Supplementary Fig. 6b).

Immunosuppressive IL-10 polarised M2c promoted Treg cell induction from allogeneic naïve CD4+ T cells in co-culture (*n* = 6 T cell monocultures, *n* = 13 co-cultures) (Fig. 6e). Co-cultures also drove enhanced Treg to immunostimulatory T effector (Teff) cell ratios (Fig. 6e). Both Treg cell induction and enhanced Treg:Teff cell ratios were restricted when M2c macrophages were FcεR:IgE cross-linked for 1 h prior to co-cultures, and abrogated when cross-linking was maintained for the 3-day co-culture (Fig. 6e). This IgE-mediated reversal of Treg cell induction was also observed in autologous co-cultures (*n* = 5)

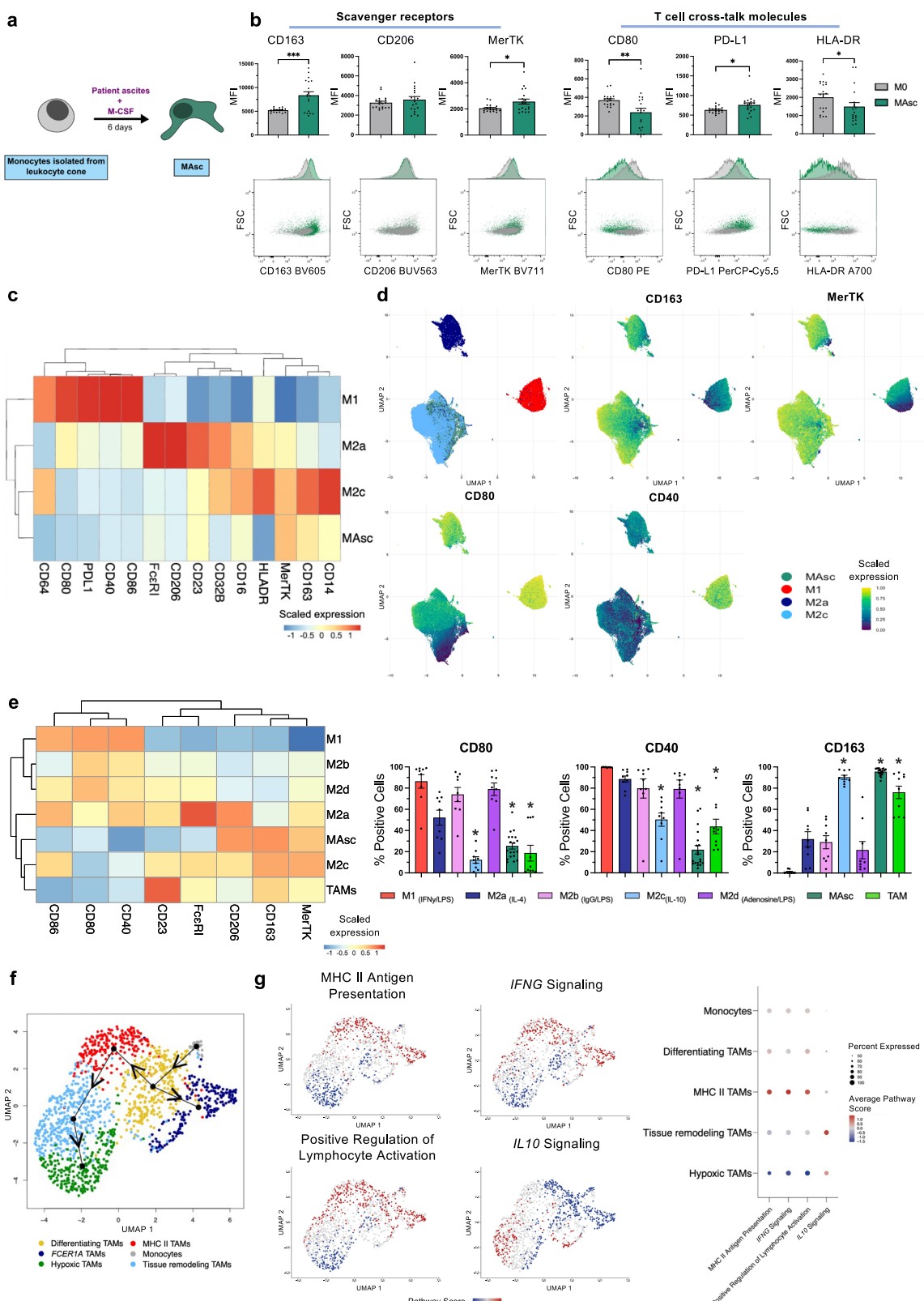

(Supplementary Fig. 6c). Additionally, FcεR:IgE cross-linking was associated with a cytokine expression shift away from immunosuppression for both Treg and Teff cells, via reduced TGF-β and IL-10 and retained TNF expression (n = 17) (Fig. 6f).

We next investigated whether this macrophage:Treg cell interaction is promoted by ovarian cancer. Similarly to M2c, MAsc, conditioned by ascites from 19 patients, promoted Treg cell induction and enhanced Treg:Teff cell ratios in co-culture with naïve CD4+ T cells, compared to both T cell monocultures and unpolarised macrophage (M0):T cell co-cultures (n = 10 T cell monocultures, n = 19 co-cultures) (Figs. 6d, 7a). Correspondingly, metastatic tumours (scRNA-seq) showed Treg enrichment relative to primary tumours (bulk RNA-seq, CIBERSORTx) and concurrent downregulation of all other CD4+ T cell lineages (T helper cells) (Supplementary Fig. 6d).

**Fig. 3 | Ovarian cancer patient ascites-conditioned macrophages (MAsc) and ascites tumour-associated macrophages (TAMs) exhibit strong phenotypic similarity to IL-10 polarised macrophages.** Evaluation of MAsc and ascites TAM phenotypes by flow cytometric assessment of surface marker expression and single-cell RNA-seq pseudotime differentiation trajectory analysis of macrophage subsets in metastatic ovarian tumours. **a** Schematic of the experimental workflow for maturing monocytes from healthy volunteer leukocyte cones into MAsc. **b** Comparison of marker expression between unpolarised M0 and MAsc, with representative flow cytometry plots (n = 19) (P = 0.0005 (CD163); 0.025 (MerTK); 0.0071 (CD80); 0.0307 (PD-L1); 0.0377 (HLA-DR)). **c** Heatmap displaying the scaled expression of markers (MFI) between MAsc (n = 19) and the in vitro-derived macrophage subsets M1, M2a and M2c (n = 6). **d** UMAP visualising MAsc (n = 19) and the in vitro-derived macrophage subsets M1, M2a and M2c (n = 6), with scaled expression of selected markers superimposed. **e** Comparison of the expression of markers across MAsc, ascites TAMs and the in vitro-derived macrophage subsets M1, M2a-d by heatmap (scaled % positive cells) and bar charts (MAsc, n = 19; TAMs,

n = 10; in vitro-derived macrophage subsets, n = 9 CD86, CD80, CD40, CD206, CD163, MerTK, FcεRI, n = 7 CD23). **f** Single-cell RNA-seq pseudotime (Slingshot package) analysis (GSE165897; n = 4 metastatic peritoneal tumours), displaying the differentiation trajectories of monocyte and macrophage subsets (n = 1374). UMAP visualising the subsets with the pseudotime minimum spanning tree super-imposed. **g** DEGs between the subsets were determined using the Seurat package (two-tailed Wilcoxon signed-rank test), and the top 100 (upregulated and down-regulated) used for gene over-representation analysis via g:Profiler (human Reac-tome (v2023.1) and Gene Ontology Biological Processes (v2023.1) databases) to investigate enrichment of gene sets. Scores for the specific statistically significant pathways between subsets visualised by UMAP on a per-cell basis and dot plot on a per-subset basis. Data are shown as mean ± SEM. Statistical significance was cal-culated using a paired two-tailed t-test (**b**) and a 1-way ANOVA with Tukey's post hoc test (**e**); (**b**) *P < 0.05, **P < 0.01 and ***P < 0.001, (**e**) *Padj <0.05. Source data and exact P/Padj values are provided as a Source Data file.

In MAsc:T cell co-cultures, 1-hour cross-linking inhibited Treg cell induction, and 3-day cross-linking abrogated Treg cell induction and the enhanced Treg:Teff cell ratio, reducing both to a similar level as T cell monocultures (Fig. 7a). Moreover, similarly to M2c co-cul-tures, MAsc enhancement of Treg and Teff cell TGF-β and IL-10 expression relative to M0, was reversed by FcεR:IgE cross-linking, whilst TNF was retained (n = 19) (Fig. 7b). Accordingly, the IgE-induced reversal of MAsc-mediated Treg cell induction and immu-nosuppressive cytokine expression was mirrored by MAsc expres-sion of TGF-β in co-culture, the master regulator of Treg cell induction (n = 12) (Fig. 7c).

We next confirmed whether IgE-mediated skewing of the ratio and phenotypes of Treg and Teff cells away from immunosuppression translated to reduced downstream T cell suppressive functions. We assessed the suppressive capacity of Treg and Teff cells, isolated from M2c:naïve CD4+ T cell co-culture, on peripheral blood mononuclear cells (PBMCs). Treg cells and Teffs exposed to IgE-repolarised M2c, were less suppressive of CD4+ and CD8+ T cell (PBMCs) proliferation, compared to those exposed to unstimulated M2c (n = 8) (Fig. 7d). This reduction in suppressive capacity was associated with higher levels of the pro-proliferative cytokine IL-2 (n = 13) (Fig. 7d).

We assessed whether this IgE-immune activation signature is of potential clinical value by stratifying patients by primary tumour *CD163* and *CD8B* expression (bulk RNA-seq) (Fig. 7e). Patients exhi-biting a *CD163*<sup>high</sup>*CD8B*<sup>low</sup> signature displayed the shortest median survival (2.98 years). However, relative to this group, patients with the *CD163*<sup>low</sup>*CD8B*<sup>high</sup> ex vivo IgE activation signature exhibited the longest median survival of all groups (5.90 years).

Finally, we probed whether this IgE-immune activation signature can be recapitulated by IgE stimulation in vivo.

Firstly, we analysed transcriptomic data from a syngeneic lung metastasis rat model, in which superior metastasis restriction was observed for rat MOv18 IgE compared to phosphate-buffered saline (PBS) control[36]. A rat tumour model was utilised as their immune sys-tem closely recapitulates the pattern of human FcεRI expression[29,37]. MOv18 IgE-treated tumours compared to control demonstrated enrichment of similar immune markers and pathways (GSEA, GOBP) as those observed in ex vivo stimulation assays (n = 2) (Fig. 8a–c). Tumours from IgE-treated animals displayed a macrophage pro-inflammatory activation signature via enhanced *TNF, CD40* and *CD80* expression and enrichment of pathways associated with Macrophage Activation, Migration and Phagocytosis and Monocyte Chemotaxis and Extravasation. Moreover, upregulated Th1 and CD8+ T cell sig-natures were observed via enrichment of *CD3E, CD8A, IFNG* and *IL12A* expression and the Positive Regulation of T cell Proliferation and *IL12* Production pathways, as well as a downregulated Treg cell signature via reduced *FOXP3* and *TGFB1* expression. Additionally, NK cell acti-vation was enriched, whilst the top 10 downregulated pathways

demonstrated diminished cell division signatures. Secondly, immu-nohistochemistry (IHC) comparison of matched pre- and on-treatment tumours of 2 ovarian cancer patients enroled in the first-in-class Phase I trial of MOv18 IgE, similarly found evidence of this IgE-immune acti-vation signature (Fig. 8d). This comprised increased intratumoural localisation of total immune cells and CD68 and CD3 expression dur-ing treatment with MOv18 IgE.

In summary, IgE-mediated induction of proinflammatory macro-phage subsets reversed patient-derived macrophage promotion of Treg cell induction and suppressive function in co-culture to increase CD8+ T cell expansion. This immune activation signature is associated with improved prognosis and is recapitulated in both tumour-bearing rats and patients following MOv18 IgE treatment.

## Discussion

We investigated ovarian cancer patient-derived macrophage pheno-types and their interactions with T cells and characterised how these can be modulated by IgE stimulation. This revealed an IgE-mediated repolarisation of patient-derived macrophage subsets away from immunosuppression towards an IgE-induced hyperinflammatory phenotype. IgE-mediated repolarisation reversed macrophage-promoted Treg cell induction and Treg cell suppressive function to increase CD8+ T cell expansion. This immunological signature was associated with improved prognosis and recapitulated in both tumour-bearing rats and patients following MOv18 IgE treatment.

We confirmed a principally immunosuppressive polarisation of ovarian cancer macrophages, associated with poor patient survival. This was determined through high-dimensional flow cytometry ana-lyses of our patient cohort (30 ascites samples) and RNA-seq analyses of publicly available, treatment-naïve primary and metastatic patient tumour cohorts. RNA-seq analysis of primary tumours identified both expression of the alternatively-activated macrophage gene *CD163* and abundance of alternatively-activated M2 macrophages to be inversely correlated with patient survival. In agreement with our findings, CD68+ CD163+ TAMs in ovarian metastases were associated with poor survival and suppressed the survival advantage provided by high expression of immune cytotoxic genes *GZMA* and *PRF1*[7]. We subse-quently probed which pathways might underpin this potentially deleterious alternative macrophage activation and found *IL10* signal-ing to be the cytokine pathway of greatest enrichment in *CD163*<sup>high</sup> primary tumours. Accordingly, TAMs from metastatic peritoneal tumours (scRNA-seq) displayed an immunosuppressive differentiation trajectory, characterised by upregulation of *IL10* signaling and down-regulation of Th1 activation pathways.

We recapitulated the immunosuppressive macrophage polarisa-tion effect of the ovarian TME ex vivo. Comparison of human macro-phages conditioned by ascites (MAsc) from 19 patients and TAMs isolated from the ascites of 10 patients, to a panel of in vitro-derived

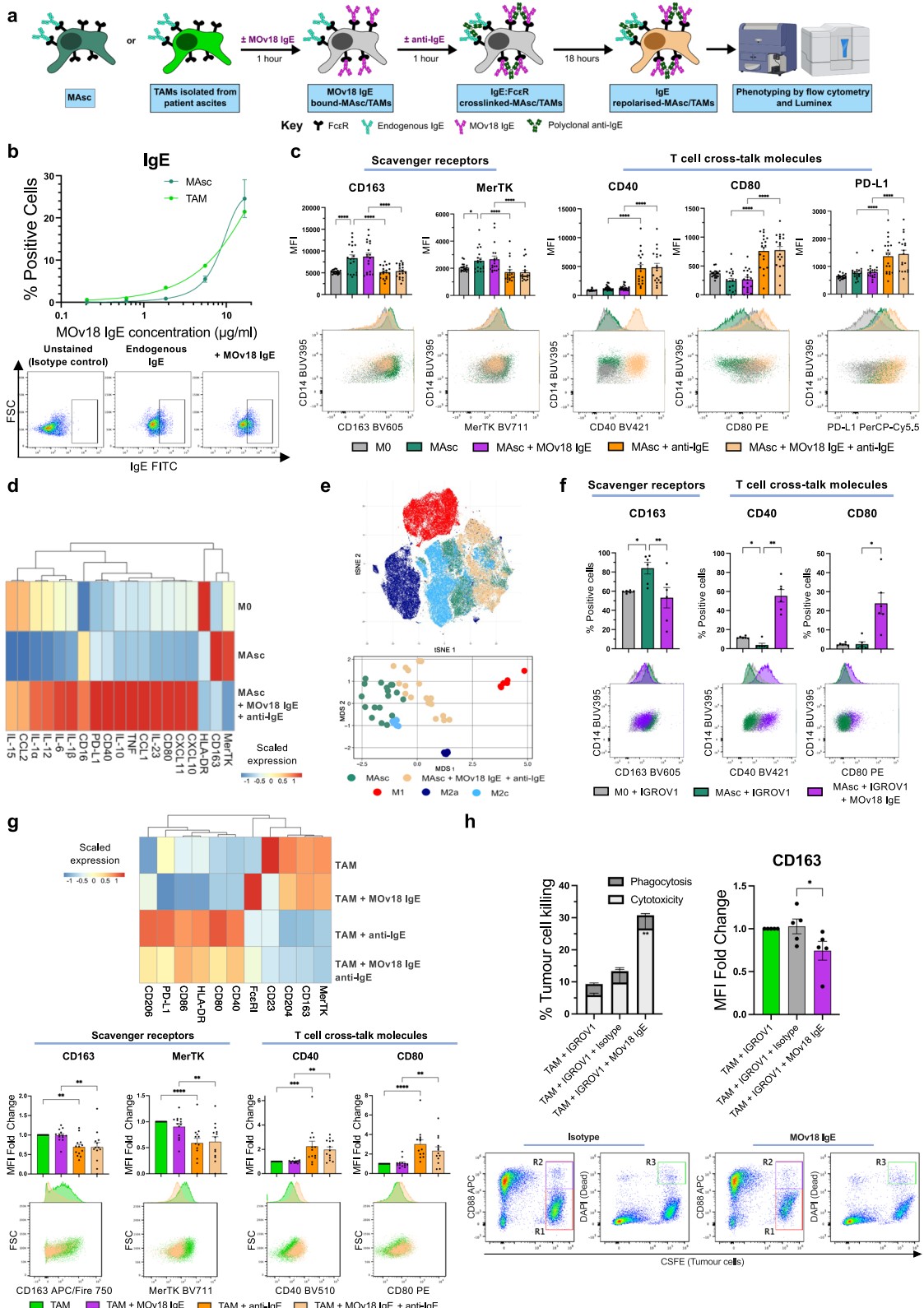

macrophage subsets, found strongest phenotypic similarity for both to immunosuppressive IL-10-polarised M2c. This correlation was driven by their expression of a CD163[high], CD40 CD80[low] signature, known to define IL-10 macrophage polarisation[38]. Although heterogeneity in ovarian TAM phenotypes is present[39–41], a conserved role for IL-10 in TAM immunosuppression has been reported across different TME sites[6,7,42,43], in accordance with our findings. In tumours, IL10+ TAMs were found to associate with poor patient survival[40] and immune

checkpoint blockade (ICB) resistance[7]. In ascites, CD163[high] CD80/86[low] ascites TAMs exhibited an IL-10[high]/IL-12[low] secretome and suppressed T cell proliferation[6]. Interestingly, treatment with IFNγ was found to reverse this immunosuppressive TAM activity[44]. Accordingly, in our study, *IFNG* signaling was downregulated in the metastatic tumour TAM differentiation trajectory, whilst classically-activated M1[high] primary tumours associated with improved patient survival. Given the known plasticity of macrophages, these pointed to the possibility that

**Fig. 4 | Ovarian cancer patient ascites-conditioned macrophages (MAsc) and ascites tumour-associated macrophages (TAMs) exhibit an IgE-mediated pro-inflammatory repolarisation and TAMs display MOv18 IgE-induced killing of ovarian cancer cells.** Evaluation of MAsc and ascites TAM MOv18 IgE binding and phenotypes following FcɛR:IgE cross-linking and MOv18 IgE-specific killing of ovarian cancer cells by TAMs. **a** Schematic of the experimental workflow for assays FcɛR:IgE cross-linking MAsc and TAM by polyclonal anti-IgE antibodies. Contains images created in BioRender. Karagiannis, S. (2025) https://BioRender.com/c06z015. **b** Assessment of MOv18 IgE binding to MAsc (n = 4) and TAMs (n = 2), with representative flow cytometry plots for TAMs. **c** Evaluation of surface marker expression on unpolarised M0 and MAsc following FcɛR:IgE cross-linking by polyclonal anti-IgE antibodies, with representative flow cytometry plots (anti-IgE: cross-linking endogenous IgE only; anti-IgE + MOv18 IgE: cross-linking endogenous IgE and exogenously applied MOv18 IgE) (n = 19). **d** Heatmap displaying the scaled expression of surface markers (MFI) (flow cytometry) and secreted factors (pg/ml) (Luminex) in cell culture supernatants, following MAsc FcɛR:IgE cross-linking by polyclonal anti-IgE antibodies (n = 19). **e** tSNE plot visualising MAsc following

FcɛR:IgE cross-linking (n = 19) and the in vitro-derived macrophage subsets M1, M2a and M2c (n = 6), based on surface marker expression (flow cytometry). MDS plot visualising the same experiment with each dot representing an individual sample. **f** Evaluation of surface marker expression on unpolarised M0 and MAsc following co-culture with FRα-expressing IGROV1 ovarian cancer cells, in the presence of MOv18 IgE, with representative flow cytometry plots (n = 6). **g** Evaluation of surface marker expression on TAMs following FcɛR:IgE cross-linking by polyclonal anti-IgE antibodies, via heatmap (scaled MFI) and bar charts, with representative flow cytometry plots (n = 13). **h** Evaluation of IGROV1 cell killing (Padj=0.0054) and TAM CD163 expression (Padj = 0.03), following co-culture between TAMs and IGROV1 cells in the presence of MOv18 IgE or isotype control (anti-NIP IgE), with representative flow cytometry plots (n = 5). Data are shown as mean ± SEM. Statistical significance was calculated using a repeated measures 1-way ANOVA with Tukey's post hoc test (**a, f, g, h left panel**) and a Friedman test with Dunn's post hoc test (**h right panel**); *Padj < 0.05, **Padj < 0.01, ***Padj < 0.001 and ****Padj < 0.0001. Source data and exact P/Padj values are provided as a Source Data file.

immunosuppressive TAM polarisation may be reversible by specific anti-tumoural stimuli.

We identified enrichment for the high-affinity IgE receptor gene, *FCER1A*, in CD163[high] primary tumours and metastatic tumour analysis demonstrated *FCER1A* expression to be highest in TAMs. Ex vivo, IgE FcɛRs were expressed at high levels by alternatively-activated M2a (IL-4-polarised) and M2c (IL-10-polarised) macrophages, and importantly in MAsc and ascites TAMs, which concordantly each displayed FcɛR enrichment in subsets with immunosuppressive phenotypes. Each of these macrophage types exhibited binding to MOv18 IgE. Given FcɛR expression in alternatively-activated macrophage subsets and immunosuppressive patient-derived macrophages, we asked whether IgE stimulation may provide an anti-tumoural modulation of TAM's immunosuppressive attributes.

We first investigated how IgE activation of macrophages is affected by polarisation state. For this we utilised the panel of in vitro-derived macrophage subsets that expands upon the binary M1/M2 polarity, which we have previously probed IgE activation of [31]. The broader panel better accounts for the spectral nature of macrophage polarisation. IL-4 polarised M2 (M2a) underwent IgE-mediated pro-inflammatory repolarisation. Importantly, the hitherto uninvestigated immunosuppressive IL-10 polarised M2c, which we demonstrated to closely resemble MAsc and TAMs, also exhibited IgE-mediated pro-inflammatory repolarisation. IFNγ/LPS-polarised M1 macrophages, which displayed negligible FcɛRI expression and MOv18 IgE binding capacity, maintained their pro-inflammatory phenotype following IgE stimulation.

In view of the ability of IgE to drive a net pro-inflammatory shift in a heterogenous population of in vitro-derived macrophage subsets, we subsequently investigated whether IgE might induce a similar repolarisation in patient-derived macrophages, away from their prevailing immunosuppressive phenotype. Following FcɛR:IgE cross-linking, both MAsc and TAMs displayed a reversal of their phenotypic similarity to M2c, through IgE-mediated CD163 and MerTK downregulation and CD40 and CD80 upregulation. This macrophage IgE activation signature could be of clinical value. For example, antibody-mediated blockade of MerTK inhibited its role in TAM efferocytosis of dying tumour cells to trigger immunogenic cell death and tumour growth restriction[45]. Moreover, IFNγ-mediated upregulation of CD80/86 on ovarian TAMs triggered enhanced tumour-specific CD8+ T cell cytotoxicity[44]. Additionally, IgE-activated MAsc upregulated checkpoint ligand PD-L1. PD-L1 is enriched in pro-inflammatory activated macrophages to limit inflammation[46] and, therefore, could act as an autoregulatory signal for IgE mAbs. This points to the potential merit of combination therapies with anti-PD-1/PD-L1 mAbs[47].

This study represents the first characterisations of patient ascites TAM engagement and repolarisation by IgE and their execution of IgE-mediated tumour cell killing via ADCC. The ability of tumour antigen-specific IgEs to mediate anti-tumoural activity has been extensively documented, culminating in a first-in-class Phase I trial for MOv18 IgE[27]. This study expands upon the previous reports of a two-armed mono-cyte/macrophage-mediated effector mechanism for IgE mAbs, comprising tumour killing and pro-inflammatory recruitment and activation[28,29,31,34,48]. In a syngeneic rat model of lung metastasis, rat MOv18 IgE induced superior tumour growth restriction compared to its IgG2b counterpart, alongside increased intra-tumoural migration of CD80+ TNF+ TAMs[29].

Our study additionally elucidates IgE-mediated repolarisation of patient-derived macrophages at a subset level. FcɛR:IgE cross-linked MAsc exhibited a shift in subsets towards pro-inflammatory activation, comprising the downregulation of 2 ascites-induced immunosuppressive subsets and a simultaneous upregulation of 3 pro-inflammatory subsets. This included an IgE-induced hyperinflammatory subset enriched for all T cell activation molecules assessed, CD40, CD80, CD86 and HLA-DR. IgE-stimulated TAMs displayed upregulation of a highly similar hyperinflammatory subset, exhibiting the identical T cell activation signature. HLA-DR enrichment in these IgE-stimulated hyperinflammatory macrophage subsets, combined with our demonstration of MOv18 IgE-specific phagocytosis of FRα-coated beads by DCs, may indicate the potential for MOv18 IgE to promote macrophage antigen presentation, as previously demonstrated for IgE in DCs[49], and warrants in-depth investigation in future studies.

Overall, these hyperinflammatory repolarisation effects indicate that IgE might reverse the TAM differentiation trajectory we observed in patient tumours, away from immunosuppressive *IL10* signaling and towards immunostimulatory interactions with anti-tumoural T cells. Consequently, we investigated how the ovarian TME influences macrophage interactions with T cells and the modulation of this axis by IgE.

We report an immunosuppressive macrophage:Treg cell interaction promoted by the ovarian TME. In metastatic tumours, Treg cells were the lymphocyte in receipt of the most TAM-derived interactions (scRNA-seq). Furthermore, enrichment of *FOXP3* expression and the Positive Regulation of Treg Differentiation pathway were observed in CD163[high] primary tumours. In co-cultures, both IL-10 polarised M2c and MAsc increased Treg cell induction from naïve CD4+ T cells and the Treg:Teff cell ratio. Additionally, MAsc enhanced Treg and Teff cell expression of immunosuppressive cytokines TGF-β and IL-10. These findings support published reports of a pro-tumoural role for Treg cell-mediated immunosuppression in ovarian cancer. Treg cells are enriched in the blood of ovarian cancer patients, and both Treg cell frequency and Treg:CD8+ T cell ratio are inversely correlated with patient survival[12,50–54]. Moreover, patient ascites Treg cells suppressed the proliferation and IFNγ secretion of autologous tumour-antigen specific CD8+ T cells[55]. With respect to the role of macrophages in promoting Treg cell-mediated immunosuppression in ovarian cancer, malignant

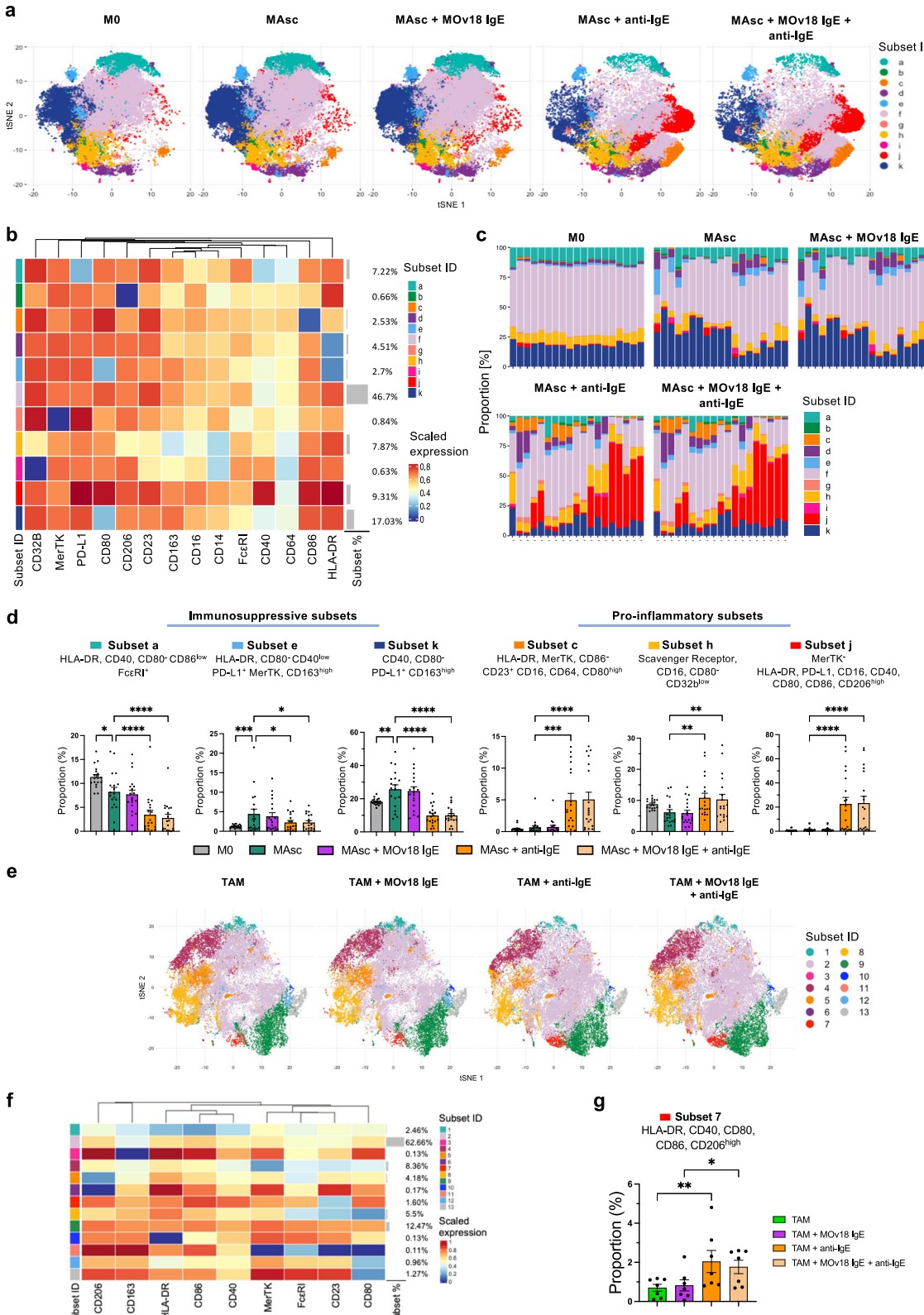

**Fig. 5 | IgE mediates a repolarisation of patient macrophage subsets away from immunosuppression towards an IgE-induced hyperinflammatory subset.**
Subset level evaluation of ovarian cancer patient ascites-conditioned macrophages (MAsc) (*n* = 19) and ascites TAMs (*n* = 7) by flow cytometric analysis of surface markers following FcɛR:IgE cross-linking by polyclonal anti-IgE antibodies. **a** tSNE plot visualising unpolarised M0 and MAsc with the 11 subsets generated by the FLOWSOM algorithm superimposed. **b** Heatmap displaying the scaled expression of markers (MFI) in each of the subsets and the mean proportion (%) of each of the subsets across the samples. **c** Comparison of the proportions (%) of each of the

subsets per sample. **d** Comparison of the proportions (%) of differentially enriched subsets. **e** tSNE plot visualising TAMs with the 13 subsets generated by the FLOWSOM algorithm superimposed. **f** Heatmap displaying the scaled expression of markers (MFI) in each of the subsets and the mean proportion (%) of each of the subsets across the samples. **g** Comparison of the proportions (%) of differentially enriched subsets. Data shown as mean ± SEM. Statistical significance was calculated using a repeated measures 1-way ANOVA with Tukey's post hoc test; *Padj <0.05, **Padj <0.01, ***Padj <0.001 and ****Padj <0.0001. Source data and exact P/ Padj values are provided as a Source Data file.

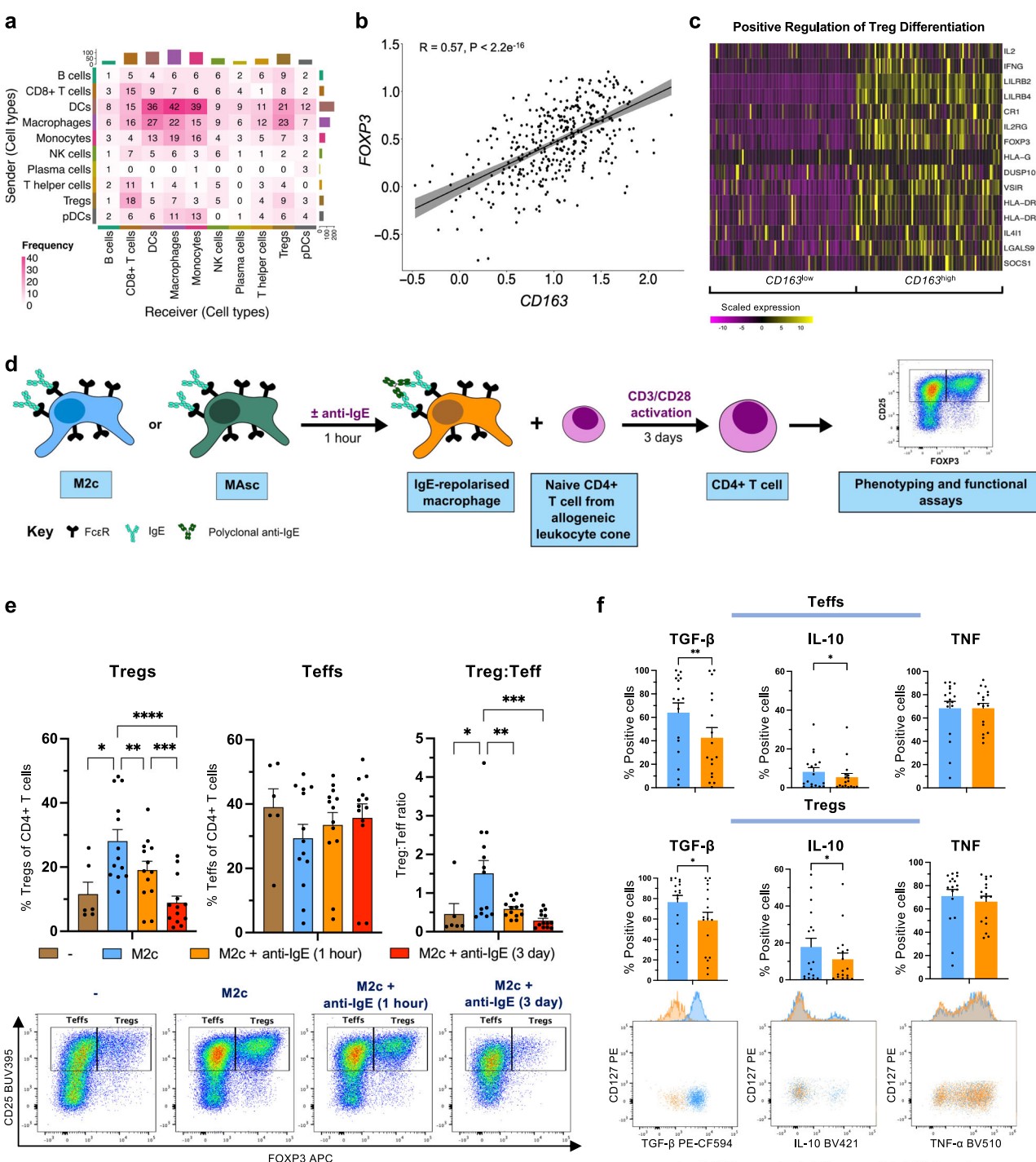

**Fig. 6 | An association between immunosuppressive macrophages and regulatory T (Treg) cells can be reversed by IgE-mediated macrophage repolarisation.** Evaluation of the association between macrophages and Treg cells in bulk (TCGA-OV; $n = 378$ primary ovarian tumours) and single-cell (GSE165897; $n = 4$ peritoneal metastatic ovarian tumours) RNA-seq datasets and ex vivo co-cultures following FcεR:IgE cross-linking (flow cytometry). **a** Evaluation of cell:cell interactions between immune cell types in GSE165897 using the Liana package. Heatmap representing the frequency of statistically significant sent and received interactions (Padj <0.01) between cell types. **b** Scatter graph displaying the Spearman's Rank Correlation between TCGA-OV tumour expression of *CD163* and *FOXP3* (R is Spearman's Rank Correlation Coefficient). **c** TCGA-OV patients were stratified by high and low levels (quartiles) of *CD163* tumour expression ($n = 190$) and differentially expressed genes (DEGs) were determined using the DESeq2 package. Gene set enrichment was evaluated within the human Gene Ontology Biological Processes (v2023.1) database by the fgsea package. Heatmap displaying the scaled expression (TPM: transcripts per million) of genes for the Positive Regulation of Treg Differentiation pathway ($n = 15$ genes). **d** Schematic of the experimental workflow for IL-10 polarised M2c and ovarian cancer patient ascites-conditioned macrophage (MAsc) co-cultures with naïve CD4+ T cells following FcεR:IgE cross-linking. **e** M2c were co-cultured with allogeneic naïve CD4+ T cells; comparison of the proportion of Treg and Teff cells of total CD4+ cells and the Treg:Teff cell ratio, with representative flow cytometry plots ($n = 6$ T cell monocultures, $n = 13$ co-cultures). **f** Comparison of the percentage of Treg and Teff cells expressing TGF-β ($P = 0.0075$ (Teff); 0.0129 (Treg)), IL-10 ($P = 0.0384$ (Teff); 0.0159 (Treg)) and TNF, with representative flow cytometry plots for Treg cells ($n = 17$). Data are shown as mean ± SEM. Statistical significance was calculated using permutation testing (**a**, **c** - gene sets), a Wald test with Benjamini−Hochberg correction (**c** - DEGs), a mixed effects analysis with Tukey's post hoc test (**e**) and a paired two-tailed $t$-test (**f**); *P/Padj < 0.05, **P/Padj < 0.01, ***P/Padj < 0.001 and **** P/Padj < 0.0001. Source data and exact P/Padj values are provided as a Source Data file.

compared to benign tumours exhibit increased frequencies of both TAMs and Treg cells, a feature associated with increased IL-10[56]. Our study significantly expands upon macrophage propagation of Treg cell activity in the ovarian TME and the specific immunosuppressive macrophage phenotypes associated with this, representing, to our knowledge, the first ex vivo demonstration of Treg cell induction by ovarian cancer patient-derived macrophages.

Importantly, we demonstrate that IgE-mediated macrophage repolarisation abrogated the macrophage:Treg cell immunosuppressive interaction, a previously unreported downstream process of IgE. FcεR:IgE cross-linking of M2c and MAsc reduced Treg cell induction, Treg:Teff cell ratios and immunosuppressive cytokine expression (TGF-β and IL-10) in Treg and Teff cells. This IgE-mediated reversal of MAsc-mediated Treg cell promotion was mirrored in MAsc TGF-β expression, the master regulator of Treg cell induction. IgE-mediated macrophage repolarisation additionally restricted the downstream immunosuppression of Treg cells on CD8+ T cell proliferation.

This ex vivo IgE-immune activation signature of immunosuppressive macrophage inhibition and T cell activation was associated with improved ovarian cancer patient prognosis and importantly was recapitulated in vivo. Firstly, MOv18 IgE-treated tumours from a syngeneic lung metastasis rat model exhibited enriched signatures of macrophage and Th1/CD8+ T cell activation and a downregulated Treg cell signature. Secondly, ovarian cancer patients from the first-in-class Phase I trial of MOv18 IgE displayed intratumoural enrichment for immune cells and CD68 and CD3 expression on-treatment. In conjunction with previous reports of increased serum levels of pro-inflammatory cytokines (IL-6, IFN-γ, CCL2) in the trial cohort following MOv18 IgE treatment and preliminary anti-tumoural activity in one patient at a low dose[27], this study further indicates that IgE may drive an in situ immune activation of potential clinical value.

In summary, through high-dimensional flow cytometric analysis of our patient cohort, integrated with public RNA-seq patient cohorts, we identified ovarian cancer macrophages to exhibit immunosuppressive phenotypes, which closely resembled IL-10 polarisation and were associated with poor patient survival. In ex vivo co-cultures, patient-derived macrophages promoted Treg cell induction and activity from naïve CD4+ T cells to collectively form an immunosuppressive axis. We demonstrated that MOv18 IgE, specific for the tumour-associated antigen FRα, can engage patient-derived macrophages ex vivo to induce killing of ovarian cancer cells and pro-inflammatory repolarisation. This repolarisation comprised a shift of macrophage subsets away from immunosuppression towards a hyperinflammatory subset enriched for T cell stimulation. Repolarisation abrogated the immunosuppressive macrophage:Treg cell axis to increase CD8+ T cell expansion, an immunological signature recapitulated in IgE-treated animals and associated with improved patient survival. On-treatment biopsies from the recent Phase I clinical trial of MOv18 IgE, which reported preliminary evidence of anti-tumoural activity[27], displayed enrichment for this IgE-immune activation signature. This study sheds light on the anti-tumoural macrophage-mediated mechanism of IgE mAbs, indicating how, in addition to tumour killing, IgE may promote a tumour immunity reorganisation beyond its described effector functions.

## Methods
### Human sample collection
The study was conducted at King's College London. Ovarian cancer patient ascites samples were collected at Guy's and St Thomas' Hospitals under a study reviewed and approved by the Guy's Research Ethics Committee (Reference 09/H0804/45), London UK. Patients were enrolled in this study by written informed consent.

The patient cohort consisted of 30 patients with epithelial ovarian cancer. Mean age: 62.7 ± 11.3 (SD); Stage: I 2, III 14, IV 14; Chemotherapy: Treated 16, Naïve 14 (Supplementary Table 1–2).

Leukocyte cones were purchased from the UK National Health Service Blood and Transplant service (NHSBT). These cones are derived from anonymised healthy volunteers following provision of written informed consent.

### Human blood cell isolation
PBMCs were isolated from healthy volunteer leukocyte cones by standard Ficoll separation, as described in Supplementary Methods.

### Patient ascites sample processing and cell isolation
Ovarian cancer patient peritoneal ascites samples were aseptically collected and processed immediately. Samples were either collected pre or peri-operatively at debulking surgery or, for palliative patients, via therapeutic drains inserted for symptom relief. Ascitic fluid was first passed through a 100 μm filter and centrifuged at 490 × g for 15 min at room temperature. For cell-free fluid collection, the supernatant was collected and passed through a 40 μm filter before a further centrifugation at 400 × g for 10 min at room temperature and storage at −80 °C.

For cell isolation, red blood cells were lysed for 5 min at room temperature and then washed. Cells were then incubated with Accutase for 10 min at room temperatures, to ensure dissociation of multicellular structures, before being passed through a 70 μm filter and washed. For mononuclear cell isolation, standard Ficoll separation was then completed.

For the isolation of TAMs, CD14+ cells were isolated from ascites mononuclear cells using CD14 MicroBeads (Miltenyi, 130-050-201), according to the manufacturer's protocol. Cells were then plated on flat-bottom cell culture plates in Macrophage Attachment Media (RPMI 1640 with 2% heat-inactivated foetal bovine serum (FBS), 2 mM L-Glutamine and 100 U/ml Penicillin-Streptomycin) at 1 million cells/ml. After a 2-h incubation (37 °C, 5% CO₂), the non-adherent cells were washed off and the media replaced with complete RPMI (RPMI 1640 with 10% FBS, 2 mM L-Glutamine and 100 U/ml Penicillin-Streptomycin).

### Human monocyte isolation from healthy volunteer leukocyte cones and ex vivo macrophage derivation
**Monocyte isolation.** Following isolation of PBMCs by Ficoll separation from leukocyte cones, monocytes were isolated by negative selection using the Pan Monocyte Isolation Kit (Miltenyi, 130-096-537), according to the manufacturer's protocol. Monocytes were then attached to flat-bottom cell culture plates using the same protocol as described above for TAM attachment.

**Macrophage derivation and polarisation.** For derivation of both in vitro-derived macrophage subsets (M0, M1, M2a-d) and MAsc, following attachment of human monocytes, complete RPMI was added, supplemented with 50 ng/ml GM-CSF (M1) or 50 ng/ml M-CSF (M0, M2a-d, MAsc) (PeproTech) (Supplementary Fig. 1a). Additionally, for MAsc, media contained 10% cell-free ascites fluid from an individual patient (Fig. 3a). Monocytes were then incubated at 37 °C, 5% CO₂.

On day 3, half of the media was replaced with complete RPMI supplemented with GM-CSF or M-CSF (50 ng/ml). For MAsc, media contained 10% cell-free ascites fluid from the same patient. MAsc derivation was complete on day 6 (Fig. 3a).

To polarise macrophages to the in vitro-derived subsets, on day 5 mature macrophages were washed and incubated for 24 h in complete RPMI with GM-CSF or M-CSF (50 ng/ml), supplemented with the following polarisation stimuli (Supplementary Fig. 1a). M0: unstimulated; M1: IFNγ (Thermo Fisher Scientific) (20 ng/ml) + LPS (Sigma–Aldrich) (100 ng/ml); M2a: IL-4 (PeproTech) (20 ng/ml); M2b: 20 μg/ml plate-bound anti-NIP ((4-hydroxy-3-iodo-5-nitrophenyl)acetic acid) IgG1 (generated in-house) + LPS (100 ng/ml); M2c: IL-10 (PeproTech) (20 ng/ml); M2d: 5′-N-Ethylcarboxamidoadenosine (NECA) (Cambridge Bioscience) (1.5 μg/ml) + LPS (100 ng/ml)

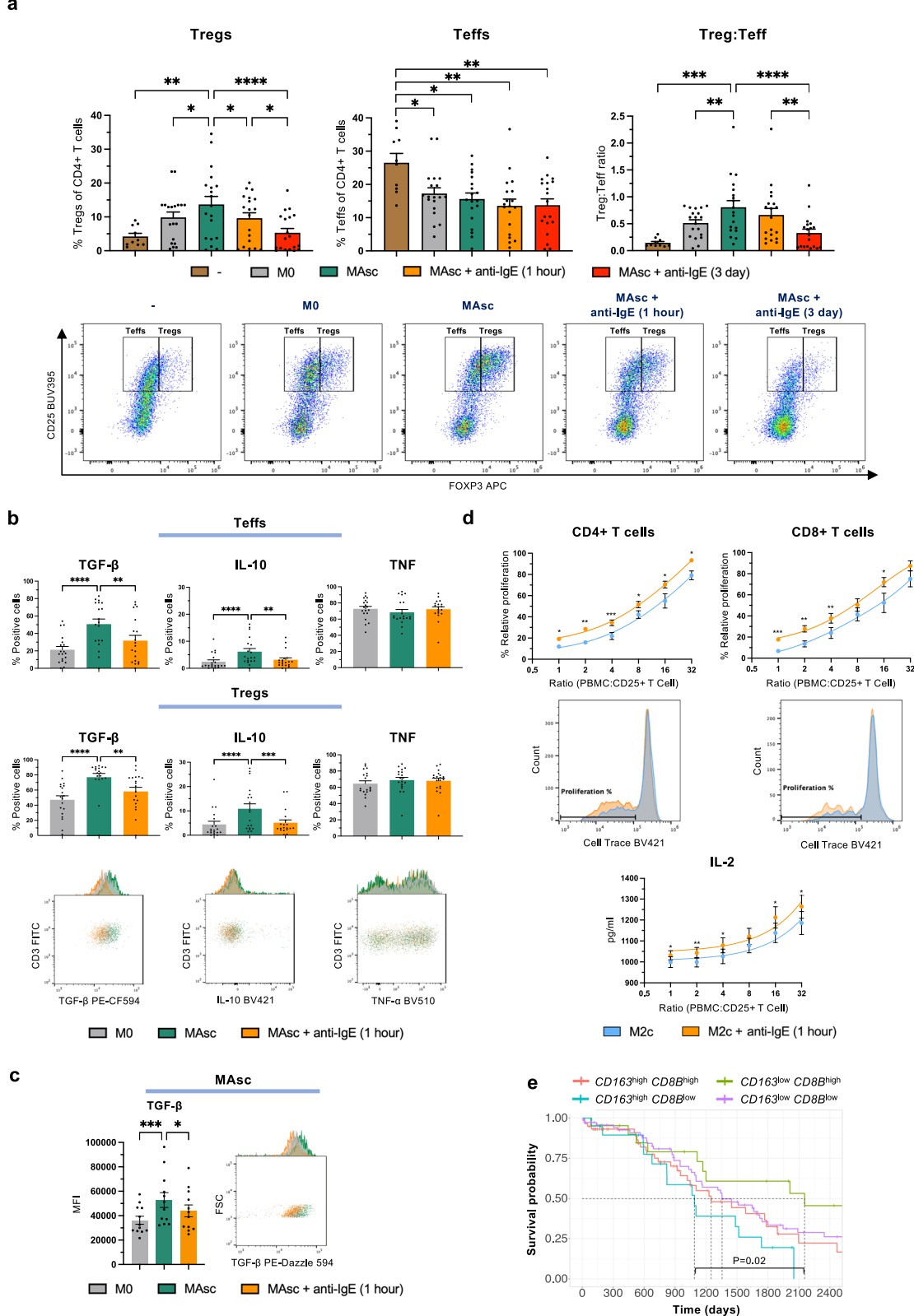

For flow cytometry experiments, detachment of macrophages (in vitro-derived subsets, MAsc and TAMs) was first completed using Accutase.

### MOv18 IgE generation

MOv18 IgE was produced from Sp2/0 hybridoma cells stably transfected with chimeric mouse/human anti-human FRα MOv18 IgE[57]. Antibody production and purification are described in Supplementary Methods.

### Macrophage FcεR:IgE cross-linking by polyclonal anti-IgE antibodies

Polyclonal anti-human IgE antibodies were used to cross-link cell surface FcεR-bound IgE on macrophages (in vitro-derived subsets, MAsc and TAMs). 10 μg/ml MOv18 IgE was added to the macrophages, followed by a 1-h incubation (37 °C, 5% $CO_2$). Cells were then washed and the media replenished with complete RPMI supplemented with 10 μg/ml polyclonal anti-human IgE (Abcam) or 10 μg/ml isotype control

**Fig. 7 | IgE-mediated hyperinflammatory repolarisation of patient macrophages reverses macrophage-promoted Treg cell activity to increase CD8+ T cell expansion.** Evaluation of effect of FcεR:IgE cross-linking in ex vivo co-cultures on: ovarian cancer patient ascites-conditioned macrophage (MAsc)-mediated Treg cell induction from allogeneic naïve CD4+ T cells and Treg cell and MAsc cytokine expression; suppressive function of Treg cells co-cultured with M2c macrophages. **a** Unpolarised M0 or MAsc were co-cultured with allogeneic naïve CD4+ T cells; comparison of the proportion of Treg and Teff cells of total CD4+ cells and Treg:Teff cell ratio, with representative flow cytometry plots ($n$ = 10 T cell monocultures; $n$ = 19 co-cultures). **b** Comparison of the percentage of Treg and Teff cells expressing TGF-β, IL-10 and TNF, with representative flow cytometry plots for Treg cells ($n$ = 19). **c** Comparison of MAsc expression of TGF-β, with a representative flow cytometry plot ($n$ = 12). **d** Following naïve CD4+ T cell co-cultures with IL-10-polarised M2c macrophages, purified CD4+ CD25+ T cells (Treg and Teff cells) were co-cultured with allogeneic peripheral blood mononuclear cells (PBMCs); comparison of the percentage relative proliferation of CD4+ and CD8+ T cells in PBMCs, with representative flow cytometry histograms (PBMC:CD25+ T cell ratio of 8) ($n$ = 8), and IL-2 concentration (pg/ml) in the cell culture supernatants (ELISA) ($n$ = 13). **e** Kaplan–Meier plot stratifying TCGA-OV patients by high and low levels (tertiles) of *CD163* and *CD8B* expression (n = 378). Data are shown as mean ± SEM. Statistical significance was calculated using a mixed effects analysis with Tukey's post hoc test (**a**), a repeated measures 1-way ANOVA with Tukey's post hoc test (**b**, **c**), a paired two-tailed $t$-test (**d**) and a Flemington-Harrington weighted log rank test (**e**); *P/Padj < 0.05, **P/Padj < 0.01, ***P/Padj < 0.001 and ****P/Padj < 0.0001. Source data and exact P/Padj values are provided as a Source Data file.

(Abcam). Following a 1-h incubation (37 °C, 5% $CO_2$), cells were washed and the media replenished with complete RPMI. Cells were then incubated (37 °C, 5% $CO_2$) for a further 18 h before evaluation of expression of cellular markers and secreted factors in culture supernatants by flow cytometry and Luminex, respectively.

## Macrophage target-specific FcεR:MOv18 IgE cross-linking by ovarian cancer cells

Macrophages (M0, MAsc and TAMs) were detached using Accutase and resuspended at $1.5 \times 10^6$ cells/ml in complete RPMI. IGROV1 ovarian cancer cells were detached using Trypsin and resuspended at $5 \times 10^5$ cells/ml. Macrophages and IGROV1 cells were co-cultured in a round bottom plate at a 3:1 ratio, with 5 μg/ml MOv18 IgE or non-specific isotype control (NIP-IgE; specific for the hapten 5-iodo-4-hydroxy-3-nitrophenyl[58]) added.

TAM-mediated killing of IGROV1 cells was evaluated after a 3-h incubation (37 °C, 5% $CO_2$), as described in Supplementary Methods, and macrophage cellular marker expression was evaluated by flow cytometry after 18 h.

Assessment of MOv18 IgE binding to macrophages and additional ex vivo IgE-stimulation assays of human basophils and DCs are described in Supplementary Methods.

## Flow cytometry staining, acquisition and analysis

Cells were placed in a round-bottom 96-well plate and washed with fluorescence-activated cell sorting (FACS) buffer (PBS with 2% FBS and 2 mM EDTA). For staining of extracellular markers, cells were resuspended in FACS buffer supplemented with fluorescently-conjugated antibodies and Fc block and incubated on ice for 30 min.

For cellular viability staining, cells were washed and resuspended in diluted viability stain. For Fixable Viability Dye (Thermo Fisher Scientific), washing and staining were completed in PBS and incubation was for 30 min, while for DAPI (4′,6-diamidino-2-phenylindole) (BD), FACS buffer was used and incubation was for 2 min.

For flow cytometry marker panels that required intracellular marker staining, the Foxp3/Transcription Factor Staining Buffer Set (eBioscience, 00-5523-00) was used, according to the manufacturer's protocol. In experiments that utilised flow cytometry panels containing more than 1 Brilliant Violet fluorophore, half of the FACS Buffer volume in the staining solution was replaced with Horizon Brilliant Stain Buffer (BD, 563794).

Samples were acquired using a BD Fortessa or Beckman Coulter CytoFLEX. All analysis was completed in FlowJo software (v10.9.0) and in RStudio using the R package CATALYST (v1.20.1).

In multi-fluorophore flow cytometry panels, Fluorescence Minus One (FMO) controls were used to gate cells expressing specific markers. In macrophage (in vitro-derived macrophages, MAsc and TAMs) flow cytometry experiments, macrophages were gated as CD14+ CD68+ (Supplementary Fig. 1b). Flow cytometry gating strategies for patient ascites cells were as follows: macrophages (CD45+ CD14+ CD88+); CD14+ DCs (CD45+ CD14+ CD88- CD1c+); CD14- DCs (CD45+ CD14- CD88- CD1c+); mast cells (CD45+ CD14- CD88- CD1c- CD117+) (Supplementary Fig. 2e); and tumour cells (EpCAM+).

For CATALYST analysis, macrophages were first gated in FlowJo, before being exported as FCS files and read into RStudio. Dimensionality reduction was completed using the tSNE and UMAP algorithms and unsupervised clustering was completed using the FLOWSOM algorithm. Heatmaps were generated, which visualised hierarchical clustering of Z-score scaled marker expression. Heatmaps were also produced using the R package pheatmap (v1.0.12).

## Luminex staining and acquisition and measurement of IgE antibodies in patient serum and ascites

Following thawing, cell culture supernatant samples were stained undiluted using custom Bio-techne Luminex kits, according to the manufacturer's protocol. Samples were acquired immediately using the Luminex FlexMap3D. IgE antibodies in ovarian cancer patient serum and ascites were measured by Synnovis (UK).

## Co-culture of macrophages with naïve CD4+ T cells

**Co-culture experimental set-up.** PBMCs from a leukocyte cone were thawed, followed by isolation of naïve CD4+ T cells using the Naïve CD4+ T Cell Isolation Kit II (Miltenyi, 130-094-131), according to the manufacturer's protocol. Isolated naïve CD4+ T cells were resuspended at $5 \times 10^5$ cells/ml in complete RPMI containing T cell TransAct (Miltenyi), a CD3/CD28 agonist, at a 1000x dilution. Macrophages (M2c or MAsc) from the same donor (autologous) or different donor (allogeneic) were cross-linked by anti-IgE (described above), washed, and then the naïve CD4+ T cells were added to the cultures. Cells were co-cultured for 72 h (37 °C, 5% $CO_2$). At the completion of the co-cultures, PBS supplemented with 2 mM EDTA was added to ensure detachment of any semi-adherent T cells.

**Flow cytometry evaluation following co-culture experiments.** Prior to staining of intracellular cytokines for flow cytometry, cells were resuspended at $1 \times 10^6$ cells/ml in T cell Media (RPMI 1640 supplemented with 10% heat inactivated Human AB serum, 2 mM L-Glutamine and 100 U/ml Penicillin-Streptomycin) supplemented with an activation cocktail comprising 50 ng/ml PMA (Phorbol 12-myristate 13-acetate) (Sigma–Aldrich), 1 μg/ml Ionomycin (Sigma–Aldrich) and 5 μg/ml Brefeldin A (Sigma–Aldrich). Cells were then incubated for 5 h (37 °C, 5% $CO_2$), followed by flow cytometry intracellular and extracellular staining.

Treg and Teff cells were gated by selecting CD4+ CD127low cells, followed by CD25high FOXP3+ (Treg) and CD25high FOXP3- (Teff) (Supplementary Fig. 6b).

**PBMC suppression assay following co-culture experiments.** Sixty hours after the initiation of co-cultures, PBMCs from an allogeneic leukocyte cone were thawed, resuspended in complete RPMI supplemented with 60 U/ml IL-2 (PeproTech) and incubated (37 °C, 5% $CO_2$) overnight. At the completion of the co-culture (72 h), these PBMCs

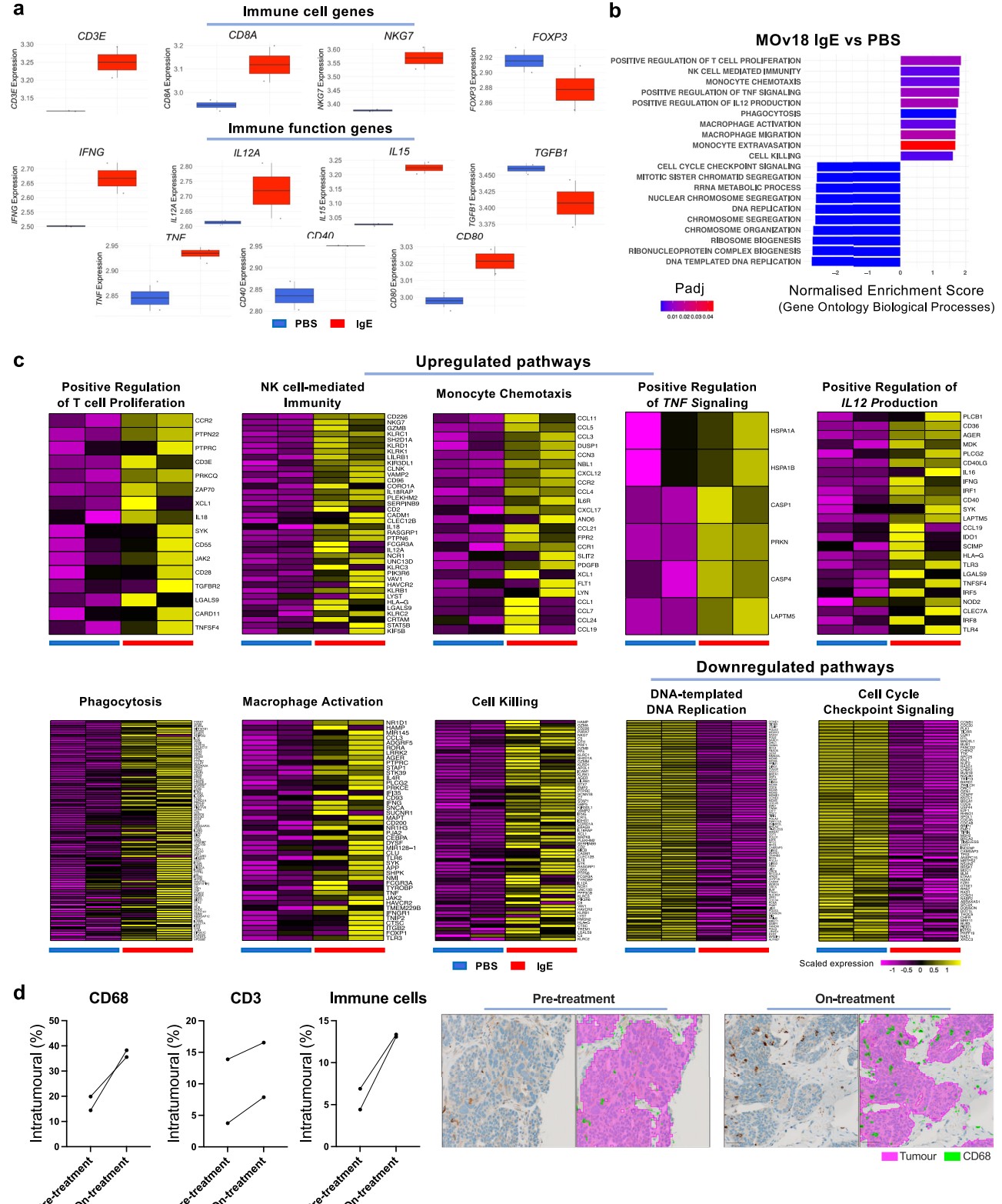

were stained with CellTrace dye (Thermo Fisher Scientific), according to the manufacturer's protocol. CellTrace dyes facilitate analysis of cellular proliferation; upon cell division the intensity of cell staining is reduced.

From the co-culture wells, CD25+ T cells (Treg and Teff cells) were isolated using CD25 MicroBeads II (Miltenyi, 130-092-983), according to the manufacturer's protocol. The isolated CD25+ T cells were resuspended at $1 \times 10^6$ cells/ml in T cell media supplemented with

500 U/ml IL-2 and transferred to a round bottom plate in 1:2 serial dilutions.

The Cell Trace-stained PBMCs were resuspended at $1 \times 10^6$ cells/ml in T cell Media supplemented with 500 U/ml IL-2 and T cell TransAct at a 400x dilution and transferred to the plate containing CD25+ T cells in serial dilutions. This resulted in a dilution series ratio of CD25+ T cells:PBMCs ranging from 1:1 to 1:32. Positive and negative control wells were prepared in triplicate, comprising PBMCs only, with and

**Fig. 8 | Following MOv18 IgE treatment, both tumours from rats and patient tumour biopsies in the Phase I trial display similar IgE-immune activation signatures to those observed in ex vivo functional assays.** Transcriptomic (microarray) evaluations of MOv18 IgE and phosphate-buffered saline (PBS)-treated tumours from a syngeneic lung metastasis rat model (n = 2) and immunohistochemistry (IHC) characterisation of matched pre- and on-treatment tumour biopsies from the Phase I trial of MOv18 IgE (n = 2). **a** Boxplot comparisons of gene expression (log₂) between tumours from MOv18 IgE- (n = 2) and PBS-treated (n = 2) rats. **b** Differentially expressed genes (DEGs) between MOv18 IgE and PBS-treated rat tumours were determined using the limma package and enrichment of genes sets evaluated within the human Gene Ontology Biological Processes (v2023.1) database by the fgsea package. Graph displays selected upregulated immune pathways and the top 10 downregulated pathways ordered by Normalised Enrichment Score. **c** Heatmaps displaying scaled expression of genes from selected upregulated and downregulated pathways: Positive Regulation of αβ T cell Proliferation (n = 16 genes); Natural Killer (NK) cell-mediated Immunity (n = 39 genes); Monocyte Chemotaxis (n = 24 genes); Positive Regulation of *TNF*-mediated signaling (n = 6 genes); Positive Regulation of *IL12* production (n = 24 genes); Phagocytosis (n = 106 genes); Macrophage Activation (n = 46 genes); Cell Killing (n = 72 genes); DNA-templated DNA Replication (n = 85 genes); Cell Cycle Checkpoint Signaling (n = 71 genes). **d** Immunohistochemical evaluation of the proportion of immune cell markers identified intratumourally in pre- and on-treatment tumour biopsies from the Phase I trial of MOv18 IgE (n = 2). Representative images of CD68 expression in pre- and on-treatment biopsies; stained images shown on the left, with the pixel classifiers used to quantify tumour (pink) and CD68 expression (green) superimposed on the right. Data shown as median (centre line), IQR (box) and range (whiskers) (**a**). Statistical significance was calculated using permutation testing (**b**). Source data and exact P/Padj values are provided as a Source Data file.

without TransAct, respectively. Cells were then incubated (37 °C, 5% CO₂) for 96 h followed by flow cytometry staining and ELISA (enzyme-linked immunosorbent assay) quantification of IL-2 (Mabtech, Human IL-2 ELISA Flex kit, 3445-1H-6) in the cell culture supernatant, according to the manufacturer's protocol. The percentage relative proliferation was determined in FlowJo by gating the proliferating cells against the negative control, followed by normalisation against the mean percentage proliferating cells of the triplicates of the positive and negative controls.

### Immunohistochemical evaluation of pre- and on-treatment tumour biopsies from the Phase I clinical trial of MOv18 IgE
IHC evaluations of paired pre- and on-treatment biopsies of metastatic tumours from two patients enroled in the Phase I clinical trial of MOv18 IgE (NCT02546921) are described in Supplementary Methods.

### In vivo analysis of MOv18 IgE-treated tumours by microarray
Preparation of the syngeneic lung metastasis rat model, treatment regimens and acquisition of Affymetrix gene chip microarray tumour data were described previously[36].

All analyses of MOv18 IgE-treated and PBS-treated tumours were completed in RStudio. Following data pre-processing, as described in Supplementary Methods, the expression matrix contained 19,058 unique transcripts (human gene names). Differential gene expression analysis was completed using the limma R package (v3.54.2) with default settings. For GSEA, all genes were ranked according to fold change and the fgsea R package (v1.22.0) was used to evaluate enrichment of gene sets within the human GOBP (v2023.1) pathway database, with default settings. Pathways with a Padj < 0.05 were considered statistically significant.

### Analysis of patient tumours (TCGA-OV) by bulk RNA-seq
TCGA-OV dataset contains bulk RNA-seq gene expression data from treatment-naïve primary tumours of high-grade serous ovarian cancer patients. Data were downloaded from Xenabrowser.net. All analysis was completed in RStudio. Following data pre-processing, as described in Supplementary Methods, the resultant dataset comprised 378 patients. Mean age: 59.6 ± 11.4 (SD); Stage: I 1; II 23; III 294; IV 57; Unknown 3. The raw count matrix contained 39,412 transcripts, and TPM matrix contained 39,430 transcripts.

Differential gene expression analysis was completed on the raw count data using the DESeq2 R package (v1.36.0) with default settings. Patients were first stratified into high and low quartiles of *CD163* expression (TPM), and DESeq2 ran. Differentially expressed genes (DEGs) with a Log2 Fold Change ≤ −1 or ≥ 1 and Padj < 0.05 were considered statistically significant. The fgsea R package was used to evaluate enrichment of gene sets within the human Reactome (v2023.1) and GOBP (v2023.1) pathway databases, as described above.

Tumour immune cell abundance was estimated from the TPM matrix using the CIBERSORTx deconvolution algorithm via the IOBR R package (v0.99.9), using default settings (CIBERSORTx LM22 signature matrix). Additional R deconvolution package ConsensusTME (v0.0.1.9000), was used to estimate overall tumour immune cell abundance (Immune Score), using default settings.

Kaplan–Meier survival analysis was completed using the R packages survival (v3.5-5) and survminer (v0.4.9), with default settings. Patients were first stratified into high and low tertiles or quartiles by individual gene expression (TPM) or outputs from the deconvolution analysis and plotted as Kaplan–Meier graphs.

### Analysis of patient tumours (GSE165897) by scRNA-seq
scRNA-seq dataset, GSE165897[35], was downloaded from Gene Expression Omnibus. Treatment-naïve, metastatic peritoneal tumours from 6 patients from GSE165897 were analysed. Mean age: 68.2 ± 4.0 (SD); Stage: III 2; IV 4. All analysis was completed in RStudio. Data pre-processing and unsupervised clustering were completed using the Seurat R package (v4.3.0.1) and are described in Supplementary Methods.

Immune cells were annotated using the following genes: T Helper cells (n = 659, genes: *CD3D⁺FOXP3⁻CD8B⁻*); Treg cells (n = 598, genes: *CD3D⁺FOXP3⁺IL2RA⁺CD8B⁻*); CD8+ T cells (n = 3078, genes: *CD3D⁺CD8B⁺TBX21⁺*); NK cells (n = 755, genes: *NKG7⁺KLRD1⁺CD8B⁻*); DCs (n = 352, genes: *LYZ⁺CLEC9A⁺C5AR1^{low}CD14^{low}*); Monocytes and Macrophages (n = 1380, genes: *LYZ⁺C5AR1⁺CD14⁺*); B cells (n = 156, genes: *MS4A1⁺CD79A⁺*); Plasma cells (n = 102, genes: *SDC1⁺CD38⁺MS4A1⁻*); Plasmacytoid Dendritic cells (pDCs) (n = 62, genes IL3RA⁺TLR7⁺) (Fig. 2e).

DEGs between the 6 monocyte and macrophage clusters (n = 1374) were determined using a minimum percentage and log fold change ≥ 0.25 and a Padj < 0.05 (Wilcoxon signed-rank test). DEGs were ranked according to Padj and the top 100 (upregulated and downregulated) were used for gene over-representation analysis via g:Profiler (https://biit.cs.ut.ee/gprofiler/gost; human Reactome (v2023.1) and GOBP (v2023.1) pathway databases; default settings) to investigate enrichment of gene sets. Pathways with a Padj < 0.05 were considered statistically significant. Subsequently clusters were annotated based on their gene (Supplementary Fig. 2c) and pathway signatures: Monocytes (n = 30); Differentiating TAMs (n = 308); *FCER1A* TAMs (n = 183); MHC II TAMs (n = 217); Tissue remodelling TAMs (n = 394); Hypoxic TAMs (n = 242).

Cell:cell interaction analysis was completed using the liana R package (v0.1.12) with default settings, to determine receptor:ligand interactions between immune cell types. A heatmap representing significant interactions between cell types (Padj < 0.01) was produced.

Pseudotime analysis was completed using slingshot R package (v2.4.0), using default settings, to map the differentiation trajectory of monocyte and macrophage subsets. Following UMAP dimensionality reduction, a minimum spanning tree was fitted to the subsets.

## Statistical analysis

Statistical analyses were performed with RStudio or GraphPad Prism (Version 10). n numbers refer to biologically independent samples only. Evaluation of data distribution was performed using the Shapiro–Wilk test and consequently, the most appropriate test for statistical significance was selected. All statistical significance tests are listed in the figure legends. Statistically significant differences are indicated in the figures as follows: $*P < 0.05$; $**P < 0.01$; $***P < 0.001$; $****P < 0.0001$.

## Reporting summary

Further information on research design is available in the Nature Portfolio Reporting Summary linked to this article.

## Data availability

The data generated in this study are provided in the Source data file. The transcriptomic datasets used in this study are publicly available: The Cancer Genome Atlas-Ovarian (TCGA-OV) can be downloaded from xenabrowser.net[59] [https://xenabrowser.net/datapages/?dataset=TCGA-OV.htseq_fpkm.tsv&host=https%3A%2F%2Fgdc.xenahubs.net&removeHub=https%3A%2F%2Fxena.treehouse.gi.ucsc.edu%3A443]; Zhang et al.[35] is available in the NCBI Gene Expression Omnibus under accession code GSE165897. All other data in the article and its Supplementary files are available from the corresponding author upon request. Source data are provided with this paper.

## Code availability

Publicly available R packages were used to conduct all analyses in RStudio, as described in Methods.

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

## Acknowledgements

G.O. is supported by the UK Medical Research Council MR/N013700/1 and is a King's College London member of the MRC Doctoral Training Partnership in Biomedical Sciences. The authors acknowledge support by the Cancer Research UK City of London Centre (C7893/A31530); the Medical Research Council (MR/V049445/1); Cancer Research UK King's Health Partners Centre at King's College London (C604/A25135); Breast Cancer Now (147; KCL-BCN-Q3); Cancer Research UK (C30122/A11527; C30122/A15774). This research was supported by the King's Health Partners Centre for Translational Medicine. The views expressed are those of the author(s) and not necessarily those of King's Health Partners. The research was supported by the National Institute for Health and Care Research Biomedical Research Centre based at Guy's and St Thomas' NHS Foundation Trust and King's College London and the NIHR Clinical Research Facility (NIHR Biomedical Research Centre at Guy's and St Thomas' NHS Foundation Trust and King's College London, IS-BRC -1215-20006). The authors are solely responsible for the study design, data collection, analysis, decision to publish, and preparation of the manuscript. The views expressed are those of the author(s) and not necessarily those of the NHS, the NIHR, or the Department of Health and Social Care. We thank all the patients and families for participating in this study. We acknowledge the Advanced Cytometry Platform at Guy's and St Thomas's NHS Foundation Trust for flow cytometry and Luminex facilities and assistance. The authors are grateful to Professor Silvana Canevari for the design of MOv18, her contribution to and critical reading of the manuscript.

## Author contributions

S.N.K. and G.O. conceived and designed the study and developed the methodology. G.O., J.L-A., R.A., M.G., R.L., G.P., M.N., A.G., H.J.B. and S.T. acquired the data or helped with the data analysis. C.S., J.H.C.L., S.G., A.M., A.S. and R.K. provided clinical support and helped to recruit patients. G.O., H.J.B., J.C., M.N., C.S., T.E. and A.S. recruited patients and acquired clinical samples. M.F., A.C., L.P., R.B. and E.J.J. provided key reagents and expertise. G.O., S.N.K., J.L-A., R.A., R.L., L.M., S.J., S.T., D.H.J. and J.S. discussed and interpreted the data. G.O. and S.N.K. wrote the manuscript. All authors edited the manuscript. S.N.K., D.H.J. and J.S. acquired funding. S.N.K. supervised the study. All listed authors agree to all manuscript contents, the author list and its order and the author contribution statements.

## Competing interests

S.N.K. and J.S. are founders and shareholders of Epsilogen Ltd. S.N.K., J.S, D.H.J., G.P. and H.J.B. declare patents on antibody technologies. H.J.B., M.G. and L.P. are funded and J.L-A. was funded via a research grant by Epsilogen Ltd. A.G., S.J. and L.M. have financial interests in SeromYx Systems. All other authors declare no conflicts of interest.

## Additional information

Gabriel Osborn[1], Jacobo López-Abente[1], Rebecca Adams[1], Roman Laddach[1,2], Melanie Grandits[1], Heather J. Bax [1], Jitesh Chauhan[1], Giulia Pellizzari [1], Mano Nakamura[1], Chara Stavraka [1,3], Alicia Chenoweth[1,4], Lais C. G. F. Palhares[1], Theodore Evan[1], Jessica Hui Cheah Lim[5], Amanda Gross[6], Lenny Moise [6], Shashi Jatiani [6], Mariangela Figini [7], Rodolfo Bianchini [8], Erika Jensen-Jarolim[8,9], Sharmistha Ghosh[5], Ana Montes[5], Ahmad Sayasneh[5], Rebecca Kristeleit [3,5], Sophia Tsoka[2], James Spicer [3], Debra H. Josephs[1,3] & Sophia N. Karagiannis [1,4] ✉

[1]St. John's Institute of Dermatology, School of Basic & Medical Biosciences & KHP Centre for Translational Medicine, King's College London, Guy's Hospital, London, UK. [2]Department of Informatics, Faculty of Natural, Mathematical and Engineering Sciences, King's College London, Bush House, London, UK. [3]School of Cancer & Pharmaceutical Sciences, King's College London, Guy's Hospital, London, UK. [4]Breast Cancer Now Research Unit, School of Cancer & Pharmaceutical Sciences, King's College London, Guy's Cancer Centre, London, UK. [5]Cancer Centre at Guy's, Guy's and St Thomas' NHS Foundation Trust, London, UK. [6]SeromYx Systems, Inc, Woburn, MA, USA. [7]ANP2, Department of Advanced Diagnostics, Fondazione IRCCS, Istituto Nazionale dei Tumori, Milan, Italy. [8]Comparative Medicine, The Interuniversity Messerli Research Institute, University of Veterinary Medicine Vienna, Medical University of Vienna, University of Vienna, Vienna, Austria. [9]Center of Pathophysiology, Infectiology and Immunology, Institute of Pathophysiology and Allergy Research, Medical University Vienna, Vienna, Austria. ✉e-mail: sophia.karagiannis@kcl.ac.uk

