## [Transparent Peer Review file · Nature Communications]

IgE induces hyperinflammatory repolarisation of ovarian cancer patient macrophages to restrict an immunosuppressive macrophage:Treg interaction

Corresponding Author: Professor Sophia Karagiannis

Version 0:

Reviewer comments:

Reviewer #1

(Remarks to the Author)

The studies from Osborn et al explore mechanisms of IgE-mediated modulation of macrophages and downstream tumour immunity in human ovarian cancer. The results point the capacity of IgE to induce a repolarized, proinflammatory state of macrophages, that inhibit Treg induction and favors Teff to Treg ratios. The focus is on MOv18, an IgE antibody that binds to tumour-associated antigen Folate Receptor- α , currently tested in the clinic. Although the results we provide useful evidence in support of IgE immunobiology in cancer treatment, the report relies heavily on in vitro work. The manuscript is well written and results support the conclusions and are well described.

Several limitations are noted:

Major.

The most informative are the results are from experiments using TAMs. However, the authors mostly focused on MAsc and M1/M2a-d polarized populations. To make it more translatable to cells found in the tumor microenvironment, can the authors address how binding of MOv18 to MAsc compares to binding to TAMs? (Or to other M2 subsets? For example comparing curves in 4g vs 1d, using same marker- percent cells or MFI-on the y axis).

With only 2 TAM cases, resemblance to results from MAsc is difficult to interpret (Fig 5). Similarly, contributions from in vivo experiments in only 2 mice (Fig 8) are modest. Additionally, a discussion is needed on how PBS injections would compare to isotype control IgE.

Importantly, there is no discussion of FR expression and whether MOv18 engages its target in vivo (Fig 8).

A discussion is needed on how the in vitro experimental conditions employed here, using cross-linking bound IgE via anti-IgE polyclonal antibody, compares to the in vivo scenario, where cross-linking would occur post-binding to FR α antigen on the tumour cell surface.

Minor

Fig 1c- The dot plots show gating strategy, with positive events in the 10e3 to 10e4 range . However, the plotted MFI numbers are 1-2 orders of magnitude lower. The authors need to specify the population (ungated?) for which MFI was plotted. MFI for Fc ϵ R1 is not indicated to be different in M1 vs M2b or M1 vs M2d. CD23 is even less striking in M1 vs M2b-d. However, in contrast to M1, binding to MOv18 shows same pattern among all M2 subsets (panel d). Is lower intensity of expression mitigated by frequency of positive cells? This could be addressed by including analyses of percent positive cells for each of the two markers, within each M population.

MHCII seems low on MAsc. It would be helpful to examine by flow whether changes in MHCII expression by MAsc and TAMs occur post MOv18 binding and cross-linking (Fig 4 f and h).

Have the authors measured the IgE found in the ascites used to generate MAsc? This is important, since some of the results suggest substantial effect from endogenous IgE, and show that exposure to exogenous IgE does not seem to trigger additional benefit.

Depending on the marker, the flow cytometry results show different numbers of clinical samples used. If multicolor flow was performed, why the lower n for some of the markers?

Reviewer #2

(Remarks to the Author)

Investigating the immuno-environment of ovarian carcinomas to understand the possible effects of anti-Folate Receptor alpha IgE (MOv18 IgE) in patients, Osborn et al report associations between IL-10-producing M2-like macrophages and Tregs. Engagement of MOv18 IgE induces repolarization of these macrophages, reversing their suppressive activity on T cells and preventing Treg induction.

The crosstalk between ovarian cancer macrophages and Treg, or the phenotype of myeloid cells in this disease, have been extensively investigated by multiple groups in the last 15-20 years. Similarly, several papers have focused on transforming myeloid cells in ovarian cancer from an immunosuppressive to an immunostimulatory cell type. The novelty of the manuscript lies on the effects of a clinically available IgE on these mechanisms. Since the pattern of expression of FcERI in humans is different from mice (as previously reported), this study could have been therapeutically relevant and potentially interesting from a mechanistic point of view. There are, however, multiple issues that should have been experimentally addressed in a different manner.

A major one is the use of artherially induced macrophages, in the absence of true TAMs. This would require the repetition of many experiments, which could not be feasible. Conditioned by ascites or not, this reviewer is not convinced that the complexity of myeloid cells in human ovarian cancer can be recapitulated using cells derived from healthy subjects, which were artificially polarized. Other FcERI+ cell types that could be relevant to understand the effects of this Ab are not included in the analyses.

MAJOR COMMENTS

-First, expression of the target of the Ab (FOLR1) in any non-tumor cell type of the ovarian cancer immune-environment, and in particular, in TAMs (true TAMs; not ascites-conditioned macrophages), must be experimentally ruled out. Human TAMs could not only express FcERI, but also low levels of folate receptor alpha. That would explain why binding of MOv18 IgE induces very different effects, compared to control IgE (e.g., in Fig 4 and Fig5). Otherwise, what is the explanation for these opposite effects?

-As aforementioned, most figures rely on artificial macrophage populations generated in vitro (including macrophages conditioned by ascites). It is irrelevant if ovarian cancer macrophages are similar to IL-10 macrophages generated in vitro or not; what matters is what is really present in the TME, plus other populations that could interact outside tumor beds with IgE. In humans, there are multiple myeloid subsets that express FcERI. Why is every dotplot specifically gated on CD14+ cells? FcERI is expressed, for instance, in human dendritic cells (as suggested by genomic data), and of course in mast cells. Expression of FcERI in these immune cells in cancer patients, plus different types of true TAMs, should be investigated by flow cytometry, and not merely by CIBERSORT; much less relying on TCGA datasets, which restricted ovarian cancer analyses to those with >70% tumor cell nuclei enrichment. The analysis of in vitro (artificially) generated macrophages must be complemented with real world samples, which are much more heterogeneous and complex.

-Binding of IgE to other cell types different from TAMs or DCs outside the TME is also relevant to understand potential toxicities in future trials. PBMCs from ovarian cancer patients should be also analyzed, if the goal is to determine the global effects of this Ab.

-Interactions between T cells and macrophages are allogeneic, which could change subtle effects in an autologous setting.

-Do hyperinflammatory TAMs induced by MOv18 IgE have the capacity to cross-present antigen (e.g., DQ-OVA)? Or merely alter cytokine production and the expression of co-stimulatory molecules?

-Figure 8 is of doubtful relevance, because the pattern of expression of FcERI in humans is different from rodents. In addition, MOv18 IgE should induce neutralizing rat Abs, which could complicate the interpretation of the results.

MINOR COMMENTS

-In the Introduction, the authors made a statement about how the affinity of FcERI for IgE should increase the half-life of IgEs, compared to IgGs. That is not factually correct, because, for instance, the half-life of Nivolumab (an IgG4 that binds very weakly to Fc Receptors) is >20 days, which appears to depend on FcRn.

Version 1:

Reviewer comments:

Reviewer #1

(Remarks to the Author)

The authors have conducted extensive additional analyses in response to the provided comments. Most notably, they have increased the patient sample size to strengthen the findings. They have also demonstrated that TAMs exhibit an immunosuppressive profile similar to MAsc, which is modulated by IgE to induce a hyperinflammatory phenotype. Additional results from clinical trial patients, showing an IgE-immune activation signature, enhance the translational relevance.

The revised manuscript includes improved flow cytometry data presentation.

While the manuscript has been significantly improved, some concerns remain, primarily related to the modest sample sizes in certain assays and inconsistencies in cohort sizes for some parameters.

Reviewer #3

(Remarks to the Author)

As a reviewer invited to assess the response of the authors to previous revisions, I carefully analyzed the authors response to reviewer 2 and I believe most of his/hers concerns were resolved in this new version of the manuscript.

General comment: The main concern of reviewer 2 was related to the artificially induced macrophages and their relevance in MOv18. In this new version authors show 1) expression of FCeR in TAM 2) MOv18 binding TAM 3) TAM repolarization to inflammatory anti-tumor profile by MOv18 binding. In my opinion, these three findings are sufficient to prove MOv18 activity on TAMs and relevance of this therapy in ovarian cancer associated macrophage polarization.

Point 1: Authors show no macrophage and high tumor cell FOLR1/FRa expression, an antigen specific activity of Mov18 mediated by macrophages. Additionally, I don't understand why Rev 2 states that MOv18-IgE and Non-tumor antigen related IgE produce opposite effects on TAMs or Masc. In figure 4 and 5 both IgE and MOv18-IgE produced similar effects on macrophages (pro-inflammatory polarization).

Point2:I believe it was correctly answered by authors by including proof of binding and polarization of macrophages to Mov18 and marginal expression of FCeR in other TME populations.

Point 3:Correctly answered by authors.

Point 4: The main goal or conclusion of this article is not about therapy preclinic or clinic toxicity, but to add evidence to their extensive literature about cellular mechanisms associated to their therapeutic tool activity. Additionally, authors already performed a Phase 1 clinical trial and toxicities are analyzed in that manuscript / assay.

Point 5: Authors included autologous co-cultures, however, is not correctly confirmed in method section that both CD4 and macrophages are isolated from the same individual. Please include that description.

Point 6: Although cross presentation by macrophages has been described, it is still an area of discussion and novel discovery. I do not believe that proving macrophage cross-presentation is mandatory to accurately show the message of this paper.

Point 7: I believe the authors correctly answered to Rev 2.

In conclusion, from the perspective of Rev 2 concerns, the authors successfully addressed all comments and major concerns.

Reviewer #4

(Remarks to the Author)

This manuscript by Osborn et al., examines the repolarisation of macrophage subsets from several sources by IgE and induction of a hyperinflammatory phenotype that restricts macrophage:Treg interactions. The monoclonal IgE antibody mOv18 (targeting folate receptor-alpha) is currently being examined in a first-in-class clinical trial. This study attempts to understand the mechanisms of IgE-mediated macrophage modulation, extending on preclinical rodent studies.

The study is highly relevant to the understanding of a IgE monoclonal antibodies for cancer immunotherapy. If proven effective in early phase trials, it will be vital to understand the immunological mechanisms unique to using IgE mAbs compared to more standard approaches and how these can be taken advantage of therapeutically. Therefore, this should be a significant piece of work of interest to experts in the field on immune-oncology.

The revised conclusions and additional work conducted, particularly on the ascites TAMs, help to validate the conclusions and findings of the manuscript. In their revised analysis the authors could place more emphasis on the ascites derived TAM studies and less of the MAsc experiments and provide additional evidence from the literature linking ascites TAM phenotype with that of intra-tumoural TAMs, as ascites samples are often used for biomarker analysis, one would assume a significant

correlation between the two sources.

Despite some caveats due to the source of macrophages used in the study, i.e., the in vitro activated or ascites induced TAMs from naïve patients vs. actual tumour infiltrating TAMs, there is still valuable fundamental information on the ability of IgE mAbs to modulate macrophage phenotype and function and instigate a hyperinflammatory phenotype. Whether this similar effect is seen in tumours of patients on treatment will need to be confirmed in later studies. The authors should take care to clarify the source of macrophages used throughout and keep consistent nomenclature. Otherwise, the experimental design and data analysis appear sound. The data is very well presented and sufficient detail of analyses and procedures is provided.

Reviewer's Comments
Point-by-point Responses

Reviewer #1 Comments:

General Comment: The studies from Osborn et al explore mechanisms of IgE-mediated modulation of macrophages and downstream tumour immunity in human ovarian cancer. The results point the capacity of IgE to induce a repolarized, proinflammatory state of macrophages, that inhibit Treg induction and favors Teff to Treg ratios. The focus is on MOv18, an IgE antibody that binds to tumour-associated antigen Folate Receptor- α , currently tested in the clinic. Although the results we provide useful evidence in support of IgE immunobiology in cancer treatment, the report relies heavily on in vitro work. The manuscript is well written and results support the conclusions and are well described.

Authors' Response

We thank the Reviewer for their positive and insightful review of our manuscript. We have sought to address all of the points raised by conducting additional experimentation and analyses, which we describe in detail in the following responses to the Reviewer's specific comments.

Major Comments:

Point 1: The most informative are the results are from experiments using TAMs. However, the authors mostly focused on MAsc and M1/M2a-d polarized populations. To make it more translatable to cells found in the tumor microenvironment, can the authors address how binding of MOv18 to MAsc compares to binding to TAMs? (Or to other M2 subsets? For example comparing curves in 4g vs 1d, using same marker- percent cells or MFI-on the y axis).

Authors' Response

We thank the Reviewer for raising this important point. We have now directly compared the binding of MOv18 IgE to ascites TAMs and MAsc via flow cytometry (please see updated Figure 4b and updated text at lines 305 and 526-528). For this we re-analysed flow cytometry experiments in which TAMs and MAsc were stained with an anti-IgE FITC polyclonal antibody. We were able to include TAMs and MAsc on the same graph in the revised manuscript, as the Reviewer suggested, by plotting the percentage of cells with an IgE signal above endogenous IgE levels at each MOv18 IgE concentration on the y-axis. This analysis demonstrated that MAsc exhibit a comparable IgE binding capacity to TAMs.

Point 2: With only 2 TAM cases, resemblance to results from MAsc is difficult to interpret (Fig 5).

Authors' Response

We thank the Reviewer for highlighting this pertinent point. To further investigate the phenotype of patient TAMs and characterise how this is modulated by IgE-mediated activation, and consequently assess how these compare to findings from MAsc, we have increased the patient number for all ascites TAM experiments and completed additional

experimental analysis (30 ovarian cancer patient ascites samples; please see updated Supplementary Tables 1-2). In our revised manuscript:

- To characterise the phenotype of TAMs, we have increased the number of patients examined from n=4 to n=10 in the revised manuscript, to confirm a principally immunosuppressive TAM phenotype which, consistent with our ascites-conditioned macrophage (MAsc) model, showed strongest phenotypic similarity to IL-10-polarised M2c macrophages (please see updated Figure 3e and updated text at lines 274-280 and 507-515). Both TAMs and MAsc expressed a CD163^{high}, CD40 CD80^{low} signature, which defines IL-10 polarisation. The additional patients analysed in the revised manuscript facilitated determination of statistical enrichment of this signature in both MAsc and TAMs, underlining the suitability of MAsc as an *in vitro* model of TAMs.
- To study IgE-mediated activation of TAMs, via *ex vivo* FcεR:IgE cross-linking using polyclonal anti-IgE antibodies, in the revised manuscript we have increased the number of patients examined from n=7 to n=13. This analysis confirmed an IgE-mediated pro-inflammatory repolarisation of TAMs away from their immunosuppressive phenotype, in accordance with IgE-activated MAsc, with both exhibiting a significant reversal of their M2c-associated CD163, MerTK^{high} CD40, CD80^{low} immunosuppressive signature (please see updated Figure 4g and updated text at lines 318-324 and 547-553).
- To characterise this IgE-mediated pro-inflammatory phenotype change at a subset level, we have increased the number of patients examined from n=2 to n=7 in the revised manuscript, which facilitated statistical analysis of differentially enriched TAM subsets. Moreover, this new analysis was completed using a flow cytometry panel with a higher number of parameters, facilitating a closer comparison to the hyperinflammatory IgE-mediated repolarisation observed for MAsc. This demonstrated an IgE-mediated upregulation of a hyperinflammatory TAM subset, which exhibited a highly similar phenotype to the hyperinflammatory subset previously observed in MAsc, with both featuring high expression of all T cell activation molecules assessed, CD40, CD80, CD86 and HLA-DR (please see updated Figure 5e-g and updated text at lines 364-369 and 573-575). This indicates the suitability of MAsc as an *in vitro* model of TAMs, suggesting that the IgE-mediated hyperinflammatory repolarisation of MAsc away from immunosuppression towards downstream T cell stimulation is also a feature of TAMs.

Point 3: Similarly, contributions from *in vivo* experiments in only 2 mice (Fig 8) are modest. Additionally, a discussion is needed on how PBS injections would compare to isotype control IgE.

Authors' Response

We thank the Reviewer for raising this important point. The transcriptomic data from the rat model of lung metastases were produced for our previously published study by Josephs et al., *Allergy*, 2018, 73(12):2328-2341 and were analysed for the purpose of this manuscript.

In our revised manuscript, we endeavoured to enhance the translational relevance of our study, and consequently now include immunohistochemical analysis of pre- and on-treatment biopsies of metastatic tumours from 2 patients enrolled in the first-in-class Phase I clinical trial of MOv18 IgE (NCT02546921) (please see new Figure 8d and new text at lines 454-458 and 617-623). In accordance with the transcriptomic data from IgE-treated rats (Figure 8a-c and text at lines 439-454 and 614-616), we identified evidence of a recapitulation of an IgE-immune activation signature in patients during MOv18 IgE treatment, through increased intratumoural localisation of immune cells and CD68 and CD3 expression in both patients.

We believe that this new analysis from the Phase I trial of MOv18 IgE, provided through our patients kindly volunteering to undergo invasive procedures, is highly relevant and appropriate for inclusion in this study. As the hypothesis of the syngeneic rat tumour model analysis was to evaluate whether the IgE-immune activation signature, identified in *ex vivo* IgE stimulation assays of patient-derived macrophages and T cells, was recapitulated in an

immunocompetent *in vivo* cancer context, the Phase I trial data compliments and significantly expands upon this; through direct assessment of the effect of MOv18 IgE on cancer patients.

We have previously demonstrated no effect on tumour growth in both ovarian cancer and melanoma mouse models engrafted with human peripheral blood mononuclear cells (PBMCs), following systemic treatment with an isotype control IgE. We include these data below for the reviewer.

Moreover, in this revised manuscript, in a co-culture between patient ascites TAMs and ovarian cancer cells, we demonstrated MOv18 IgE-specific killing of cancer cells by TAMs, with an isotype control IgE having no effect compared to untreated co-cultures (please see new Figure 4h and new text at lines 329-336 and 558-560).

[panel redacted]

Figure 1: Non-specific isotype control IgE has no effect on *in vivo* tumour model.

(a) SCID mice were challenged in the flank with FR α + IGROV1 ovarian cancer cells and injected with healthy volunteer-derived PBMCs and antibodies. Mice treated with anti-FR α MOv18 IgE had significantly lower tumour sizes, compared to those treated with non-specific isotype control IgE (SF-25 IgE) (Taken from Gould et al., *Eur J Immunol*, 1999, 29(11):3527-37, Figure 4b). (b) Subcutaneous melanoma (CSPG4+ A375 melanoma cells) NSG mouse model engrafted with melanoma patient-derived PBMCs. Upper: design and dosing regimen for mouse model. SC = subcutaneous. Lower left: mice treated with anti-CSPG4 IgE had significantly lower tumor weights at the end of the study, compared to those treated with non-specific isotype control IgE or PBS. Lower right: In the same model, tumor volume was significantly lower in animals treated with anti-CSPG4 IgE, compared to non-specific isotype control IgE or PBS. Inset graphs to the right show tumor growth curves for individual animals. Taken from Chauhan et al, *Nat Commun*, 2023, 14(1):2192, Figure 7b).

Point 4: Importantly, there is no discussion of FR expression and whether MOv18 engages its target *in vivo* (Fig 8).

Authors' Response

Thank you for raising the need for this clarification. This rat syngeneic lung metastases model has been characterised in previous publications by our group (Josephs et al., *Cancer Res*, 2017, 77(5):1127-1141 & *Allergy*, 2018, 73(12):2328-2341). The tumours comprise the syngeneic WAG rat adenocarcinoma cell line CC531 transfected with human FR α (CC531tFR), which form lung metastases when injected systemically. Rats were dosed with PBS or a mouse/rat chimeric MOv18 IgE (rMOv18 IgE). We previously demonstrated binding of rMOv18 IgE to FR α -expressing CC531tFR but not to FR α - tumour cells (Figure 2a), as well as to rat primary monocytes (Figure 2a), which express the same trimeric form of Fc ϵ RI as human monocytes/macrophages (Figure 2b). Consequently, rat primary monocytes mediated *ex vivo* FR α -specific killing of CC531tFR by rMOv18 IgE, via ADCC, as determined by the inclusion of a rat isotype control IgE (Figure 2c). We include these data below for the Reviewer.

Editorial Note: full citation for the above figure 1b: Chauhan, J., Grandits, M., Palhares, L.C.G.F. et al. Anti-cancer pro-inflammatory effects of an IgE antibody targeting the melanoma-associated antigen chondroitin sulfate proteoglycan 4. *Nat Commun* 14, 2192 (2023). <https://doi.org/10.1038/s41467-023-37811-3>

[figure redacted]

Figure 2: Rat MOv18 IgE binds to rat tumour cells expressing human FR α and rat primary monocytes and induces FR α -specific tumour cell killing by rat monocytes.

(a) Rat MOv18 IgE binds to rat primary monocytes and rat adenocarcinoma cells expressing human FR α (CC531tFR) but not to FR α - cancer cells (grey - nonspecific isotype control; white - detection antibody). **(b)** Rat primary CD172+ monocytes express Fc ϵ RI. **(c)** Rat MOv18 IgE mediates FR α -specific killing of CC531tFR by rat primary monocytes, via ADCC (taken from Josephs et al., *Cancer Res*, 2017, 77(5):1127-1141, Figure 1a-c).

Furthermore, we also previously demonstrated that when administered to the rat model, MOv18 IgE promotes: a) macrophage infiltration from the tumour periphery to tumour islets (Figure 3a); b) restriction of tumour growth (Figure 3b); and c) reduction of FR α + tumour cells (Figure 3a). We include these data below for the Reviewer.

[figure redacted]

Figure 3: MOv18 IgE treatment induces macrophage tumour infiltration and restriction of tumour growth in a syngeneic lung metastases rat model.

An immunocompetent lung metastases rat model, produced through systemic injection of the syngeneic WAG rat adenocarcinoma cell line CC531 transfected with human FR α , treated with mouse/rat chimeric MOv18 IgE, mouse/rat chimeric MOv18 IgG or PBS control. **(a)** Paraffin-embedded lung sections show increased CD68+ macrophages and reduced FR α + tumour cells in tumour islets, following MOv18 IgE treatment compared to MOv18 IgG and PBS control. **(b)** MOv18 IgE treatment is associated with increased lung metastasis reduction

compared to MOv18 IgG and PBS control (taken from Josephs et al., *Cancer Res*, 2017, 77(5):1127-1141, Figure 2c, 3c).

These data are not only consistent with findings in this study from tumour biopsies of patients enrolled in the Phase I trial of MOv18 IgE, in which we demonstrated increased intratumoural CD68 expression on-treatment compared to pre-treatment, but also indicate the elimination of FR α + tumour cells *in vivo*.

To further evaluate the capacity of TAMs to engage with and eliminate FR α + tumour cells in our revised manuscript, we conducted further experiments in which we demonstrated:

- High expression of *FOLR1*/FR α on tumour cells from ovarian cancer patients, via scRNA-seq analysis of metastatic peritoneal tumours and flow cytometry evaluation of ascites cells (please see new Supplementary Figure 4b-c and next text at lines 330-333).
- MOv18 IgE-specific killing of FR α -expressing human ovarian cancer cells by TAMs *ex vivo*, using an IgE isotype control to ensure antibody target specificity (please see new Figure 4h and new text at lines 329-336 and 558-560).

Point 5: A discussion is needed on how the *in vitro* experimental conditions employed here, using cross-linking bound IgE via anti-IgE polyclonal antibody, compares to the *in vivo* scenario, where cross-linking would occur post-binding to FR α antigen on the tumour cell surface.

Authors' Response

Thank you for raising this point. To address how IgE:Fc ϵ R cross-linking using polyclonal anti-IgE antibodies compares to specific MOv18 IgE:Fc ϵ R cross-linking on TAMs, in the revised manuscript we include co-culture assays between ascites TAMs and FR α + IGROV1 ovarian cancer cells (please see new Figure 4h and new text at lines 329-336 and 558-560). MOv18 IgE-specific Fc ϵ R cross-linking via FR α , as determined by the inclusion of an isotype control IgE, triggered TAM activation. This comprised: a) ADCC-mediated killing of IGROV1s; and b) TAM CD163 downregulation - a finding consistent with our data showing pro-inflammatory activation of TAMs following *ex vivo* treatment with polyclonal anti-IgE to cross-link all Fc ϵ R-bound IgE (please see updated Figure 4g and updated text at lines 318-324 and 547-553). These findings indicate that the IgE-mediated pro-inflammatory macrophage activation effects can be recapitulated in an antigen-specific setting. Furthermore, macrophage activation in response to MOv18 IgE was also indicated in on-treatment tumour biopsies from patients enrolled in the Phase I trial, through increased intratumoural CD68 expression (please see new Figure 8d and updated text at lines 454-458 and 617-623).

Minor Comments:

Minor Point 1: Fig 1c- The dot plots show gating strategy, with positive events in the 10e3 to 10e4 range . However, the plotted MFI numbers are 1-2 orders of magnitude lower. The authors need to specify the population (ungated?) for which MFI was plotted. MFI for Fc ϵ R1 is not indicated to be different in M1 vs M2b or M1vs M2d. CD23 is even less striking in M1 vs M2b-d. However, in contrast to M1, binding to MOv18 shows same pattern among all M2 subsets (panel d). Is lower intensity of expression mitigated by frequency of positive cells? This could be addressed by including analyses of percent positive cells for each of the two markers, within each M population.

Authors' Response

The MFI values plotted in the bar charts in Figure 1c were calculated by taking the ungated macrophage Fc ϵ R1 and CD23 MFI values and subtracting the MFI from the fluorescence minus one (FMO) control. Consequently, this resulted in the plotted MFI values having lower

values than the MFI of the FcεRI+ and CD23+ macrophage populations shown on the flow cytometry plots, as correctly pointed out by the Reviewer.

To provide a more accurate representation of FcεRI and CD23 expression across the macrophage subsets, we have followed the Reviewer's advice and presented the bar charts as percentage positive cells (please see updated Figure 1c and text at lines 182-185 and 524-531). This additionally addresses the second point raised by the Reviewer with respect to whether M2b and M2d show higher FcεRI and CD23 expression compared to M1, for which negligible expression is present. In the revised figure, M2b and M2d consistently displayed FcεRI+ and CD23+ cells across donors, whilst M1 still showed negligible expression.

These findings are in accordance with the MOv18 IgE binding data presented in Figure 1d, in which M1 showed negligible binding, while all M2 subsets, including M2b and M2d, showed binding to MOv18 IgE at a similar level. The use of an anti-IgE FITC polyclonal antibody in this assay to stain for MOv18 IgE binding limits the ability to distinguish differential MOv18 IgE capacities within the M2 subsets and assess whether these correspond to the relatively small differences in FcεR expression observed between the M2 subsets in Figure 1c. This is because a polyclonal antibody will bind multivalently to the FcεR-engaged MOv18 IgE to amplify the fluorescent signal, reducing the resolution in determining differential binding. However, use of this methodology ensures the negligible binding of MOv18 IgE to M1 represents a true negative. Accordingly, we only conclude in the manuscript that all subsets apart from M1 displayed binding to MOv18 IgE.

Minor Point 2: MHCII seems low on MAsc. It would be helpful to examine by flow whether changes in MHCII expression by MAsc and TAMs occur post MOv18 binding and cross-linking (Fig 4 f and h).

Authors' Response

In the revised manuscript we have now included flow cytometric evaluation of HLA-DR expression on MAsc and on ascites TAMs in basal conditions and following FcεR:IgE cross-linking using polyclonal anti-IgE antibodies. As mentioned by the Reviewer, HLA-DR was downregulated by ascites conditioning, and this lower expression was unaffected by IgE-mediated MAsc activation at a population level (please see new Supplementary Figure 3c and new text at lines 311-312). IgE-activated TAMs displayed a trend towards HLA-DR upregulation at a population level, however this did not reach statistical significance (please see new Supplementary Figure 3d and new text at lines 322-324).

In the revised manuscript, we have updated the subset level characterisation of IgE-activated ascites TAMs, having increased the number of patients examined from n=2 to n=7; facilitating statistical analysis of differentially enriched TAM subsets. Consistent with subset level analysis of MAsc included in the original manuscript, we show that both cell types displayed an IgE-mediated upregulation of a hyperinflammatory subset, with each exhibiting high expression of all T cell activation molecules assessed, including HLA-DR (Subset *j* for MAsc and Subset 7 for TAMs) (please see Figure 5a-d and text at lines 348-363 and updated Figure 5e-g and updated text at 364-369, respectively, and updated Discussion text at lines 568-579).

Minor Point 3: Have the authors measured the IgE found the ascites used to generate MAsc? This is important, since some of the results suggest substantial effect from endogenous IgE, and show that exposure to exogenous IgE does not seem to trigger additional benefit.

Authors' Response

In the revised manuscript we have conducted additional analyses to measure the concentrations of IgE antibodies in ovarian cancer patient serum (n=15) and ascites (n=20). We found IgE to be consistently detectable in both serum and ascites, with lower levels found in ascites (please see new Supplementary Figure 4a and new text at lines 324-328). Previous

analysis from our group has found consistent serum IgE levels in patients compared to healthy volunteers (Nakamura et al., *Cancers*, 2020, 12(11):3376).

The FcεRI:IgE interaction is the highest affinity interaction of any antibody isotype for an Fc receptor; consequently IgE is the only antibody isotype strongly retained by immune effector cells in the absence of antigen engagement (McDonnell et al., *Nat Struct Biol*, 2001, 8(5):437-41). Therefore, FcεRI-bound IgE is retained on human immune cells following isolation, as well after *ex vivo* culture on macrophages as we previously demonstrated (Pellizzari et al., *EBioMedicine*, 2019, 43:67-81). As a result, TAMs retain FcεRI-bound IgE from ascites following *ex vivo* culturing, whilst MAsc can retain FcεRI-bound IgE from healthy volunteer blood following monocyte isolation and from patient ascites following *ex vivo* ascites conditioning. Consequently, a comparable activation effect is as expected observed for both MAsc and TAMs when cross-linking endogenous IgE only compared to both endogenous and exogenous IgE, as only a fraction of occupied FcεRIs need to be cross-linked to achieve a maximal effect; consistent with previous analyses from our group on human macrophages and monocytes (Pellizzari et al., *EBioMedicine*, 2019, 43:67-81; Chauhan et al., *Nat Commun*, 2023, 14(1):2192). Accordingly, in the revised manuscript, we have shown that when only FcεRs occupied by MOv18 IgE are cross-linked by engagement of FRα expressed on IGROV1 ovarian cancer cells, CD163 is downregulated on TAMs to a similar degree as with anti-IgE-mediated cross-linking (please see updated Figure 4g and updated text at lines 318-324 and new Figure 4h and new text at lines 329-336, respectively).

Minor Point 4: Depending on the marker, the flow cytometry results show different numbers of clinical samples used. If multicolor flow was performed, why the lower n for some of the markers?

Authors' Response

Thank you for raising the need to clarify this point. In the evaluation of the IgE-mediated activation of *in vitro*-derived macrophage subsets in Figure 1e, different flow cytometry panels were used, resulting in a different n number depending on the marker. Scavenger receptors were stained for all donors (n=9), whereas FcεRI (n=6) and T cell cross-talk molecules (n=5) were not stained for all donors.

However, for the characterisation of MAsc and ascites TAMs, a consistent flow cytometry panel was used for each cell type, resulting in the same n number for all markers in each experiment. We clarify the n numbers for all analyses in the main manuscript and these are also specified in the figure legends.

Reviewer #2 Comments:

General Comment: Investigating the immuno-environment of ovarian carcinomas to understand the possible effects of anti-Folate Receptor alpha IgE (MOv18 IgE) in patients, Osborn et al report associations between IL-10-producing M2-like macrophages and Tregs. Engagement of MOv18 IgE induces repolarization of these macrophages, reversing their suppressive activity on T cells and preventing Treg induction.

The crosstalk between ovarian cancer macrophages and Treg, or the phenotype of myeloid cells in this disease, have been extensively investigated by multiple groups in the last 15-20 years. Similarly, several papers have focused on transforming myeloid cells in ovarian cancer from an immunosuppressive to an immunostimulatory cell type. The novelty of the manuscript lies on the effects of a clinically available IgE on these mechanisms. Since the pattern of expression of FcERI in humans is different from mice (as previously reported), this study could have been therapeutically relevant and potentially interesting from a mechanistic point of view. There are, however, multiple issues that should have been experimentally addressed in a different manner.

A major one is the use of arterially induced macrophages, in the absence of true TAMs. This would require the repetition of many experiments, which could not be feasible. Conditioned by ascites or not, this reviewer is not convinced that the complexity of myeloid cells in human ovarian cancer can be recapitulated using cells derived from healthy subjects, which were artificially polarized. Other FcERI+ cell types that could be relevant to understand the effects of this Ab are not included in the analyses.

Authors' Response

We thank the Reviewer for their detailed and insightful review of our manuscript. The focus of our study is on the effects of the clinically evaluated MOv18 IgE on macrophages from patients with epithelial ovarian cancers. Our work builds upon extensive pre-clinical evidence of the anti-tumoural efficacy for IgE mAbs, via macrophage-mediated Fc functions, in several published studies (Karagiannis et al., *J Immunol*, 2007, 179(5):2832-43; Josephs et al., *Cancer Res*, 2017, 77(5):1127-1141 & *Allergy*, 2018, 73(12):2328-2341; Pellizzari et al., *EBioMedicine*, 2019, 43:67-81; Chauhan et al, *Nat Commun*, 2023, 14(1):2192).

Here, we characterise IgE-mediated engagement and activation of ovarian cancer patient macrophages, via high-dimensional flow cytometry analysis of patient ascites and *ex vivo* functional studies, integrated with analysis of, transcriptomic datasets from treatment-naïve patient tumours, FR α -expressing tumour-bearing rats and now patients from the Phase I trial of MOv18 IgE.

In our revised manuscript:

We have further characterised ovarian cancer patient tumour-associated macrophage (TAMs) and we have assessed how they are modulated by IgE-mediated activation. We have increased the patient sample numbers for all ascites TAMs experiments across different parts of our study. We have completed new experiments on TAMs to characterise their capacity to engage MOv18 IgE and undergo MOv18 IgE-specific activation and killing of FR α -expressing ovarian cancer cells. Furthermore, we now include immunohistochemical analysis of pre- and on-treatment tumour biopsies from two patients enrolled in the Phase I trial of MOv18 IgE, in which we identified increased intratumoural localisation of immune cells and CD68 and CD3 expression during treatment. Collectively, these new and updated analyses enhance the clinical relevance of the findings of our study.

We have sought to address all of the points raised by conducting additional experimentation and analyses, which we describe in detail in the following responses to the Reviewer's specific comments.

Major Points:

Point 1: First, expression of the target of the Ab (FOLR1) in any non-tumor cell type of the ovarian cancer immune-environment, and in particular, in TAMs (true TAMs; not ascites-conditioned macrophages), must be experimentally ruled out. Human TAMs could not only express FcERI, but also low levels of folate receptor alpha. That would explain why binding of MOv18 IgE induces very different effects, compared to control IgE (e.g., in Fig 4 and Fig5). Otherwise, what is the explanation for these opposite effects?

Authors' Response

We thank the Reviewer for raising this important point. In our revised manuscript:

- We characterise *FOLR1/FR α* expression on TAMs and ovarian tumour cells at both an RNA and protein level, via scRNA-seq analysis of the GSE165897 dataset of metastatic peritoneal tumours (please see new Supplementary Figure 4b and new text at lines 330-333) and flow cytometric analysis of patient ascites (please see new Supplementary Figure 4c and new text at lines 330-333), respectively. These clearly demonstrated:
 - Ovarian cancer TAMs (from patients): TAMs showed no RNA or protein expression of *FOLR1/FR α* .
 - Ovarian cancer tumour cells (from patients): Tumour cells showed high RNA and protein expression of *FOLR1/FR α* .
- Consistent with the lack of *FR α* expression on macrophages, throughout our study we have found MOv18 IgE treatment alone, in the absence of cross-linking via polyclonal anti-IgE antibodies or via crosslinking of the antigen *FR α* , to have no effect on macrophages, observed in: a) healthy volunteer-derived macrophage subsets (please see Figure 1e and text at lines 187-200); b) ascites-conditioned MAsc (please see Figure 4c and text at lines 306-312) and; c) ascites TAMs (please see updated Figure 4g and updated text at lines 318-324).
- We also demonstrate that MOv18 IgE-specific cross-linking of ascites TAMs by human ovarian cancer cells expressing the target antigen, Folate Receptor- α (*FR α*), but not by the non-specific IgE isotype control, mediated antibody-dependent cellular cytotoxicity (ADCC) killing. This also resulted in downregulation of the anti-inflammatory scavenger receptor CD163 (please see new Figure 4h and new text at lines 329-336 and 558-560). These demonstrate the effects of IgE stimulation in an antigen-specific manner.

Point 2: As aforementioned, most figures rely on artificial macrophage populations generated in vitro (including macrophages conditioned by ascites). It is irrelevant if ovarian cancer macrophages are similar to IL-10 macrophages generated in vitro or not; what matters is what is really present in the TME, plus other populations that could interact outside tumor beds with IgE.

Authors' Response

We thank the Reviewer for highlighting this important point. In the revised manuscript we have extensively broadened our characterisation of the phenotype of TAMs from the ovarian tumour microenvironment (TME). We have increased the patient number for all ascites TAM experiments and performed new experimental analysis (30 ovarian cancer patient ascites samples; please see updated Supplementary Tables 1-2).

This was completed by flow cytometric evaluation of TAMs from patient ascites, via a flow cytometry panel with a higher number of parameters than used in the original manuscript. This extended the characterisation of the phenotype of TAMs, in basal conditions and following IgE stimulation.

- With respect to characterising the phenotype of TAMs, in the revised manuscript we have increased the number of patient samples examined from n=4 to n=10 to confirm a principally immunosuppressive TAM phenotype which, consistent with our ascites-conditioned macrophage (MAsc) model, showed strongest phenotypic similarity to IL-10-polarised M2c macrophages (please see updated Figure 3e and updated text at lines 274-280 and 507-515). Both TAMs and MAsc expressed a CD163^{high}, CD40 CD80^{low} signature, which defines IL-10 polarisation. The additional TAM patients analysed in the revised manuscript facilitated determination of statistical enrichment of this signature in both MAsc and TAMs, underlining the suitability of MAsc as an *in vitro* model of TAMs.
- Additionally, we include a new evaluation of TAM FcεRI and CD23 expression in samples from 14 patients, which demonstrated consistent expression of both IgE receptors at a protein level (please see new Figure 2g, top, and new Supplementary Figure 2d and new text at lines 246-248 and 525-528). This is in accordance with the scRNA-seq analysis of immune cells from metastatic tumours from the same intraperitoneal space as ascites, which identified a FCER1A+ TAM subset consistently across patients (please see Figure 2f and text at lines 242-245).
- With regards to the characterisation of IgE-mediated activation of TAMs, via *ex vivo* FcεR:IgE cross-linking using polyclonal anti-IgE antibodies, in the revised manuscript we have increased the number of patients examined from n=7 to n=13. This confirmed an IgE-mediated pro-inflammatory repolarisation of TAMs, consistent with IgE-activated MAsc, with both exhibiting a significant reversal of their M2c-associated CD163, MerTK^{high} CD40, CD80^{low} immunosuppressive signature (please see updated Figure 4g and updated text at lines 318-324 and 547-553).
- With respect to the characterisation of this IgE-mediated pro-inflammatory phenotype change at a subset level, we have increased the number of patients examined from n=2 to n=7; this facilitated statistical analysis of differentially modulated TAM subsets. Consequently, we identified an IgE-mediated upregulation of a hyperinflammatory subset, exhibiting a highly similar phenotype to the IgE-induced hyperinflammatory subset previously observed in MAsc, comprising high expression of T cell activation molecules assessed (CD40, CD80, CD86 and HLA-DR) (please see updated Figure 5e-g and updated text at lines 364-369 and 573-575).
- Additionally, we have completed new experiments on TAMs to characterise their capacity to engage MOv18 IgE and undergo MOv18 IgE-specific activation:
 - Characterised MOv18 IgE-specific cross-linking of ascites TAMs by human IGROV1 ovarian cancer cells expressing the target antigen, Folate Receptor-α (FRα), compared with non-specific isotype control. This demonstrated that TAMs mediate ADCC killing of ovarian cancer cells by MOv18 IgE in an antigen-specific manner. This was associated with a downregulation of anti-inflammatory scavenger receptor CD163 (please see new Figure 4h and new text at lines 329-336 and 558-560), consistent with the pro-inflammatory TAM repolarisation induced by anti-IgE-mediated FcεR:IgE cross-linking (please see updated Figure 4g and updated text at lines 318-324 and 547-553).
 - Additionally, we now include immunohistochemical analysis of pre- and on-treatment tumour biopsies from 2 patients enrolled in the Phase I clinical trial of MOv18 IgE, in which we identified increased intratumoural TAM infiltration during treatment, as determined by staining of CD68 (please see new Figure 8d and new text at lines 364-369 and 617-623).

In summary, we have significantly enhanced the clinical relevance of our work by updated and additional phenotypic and functional characterisations of patient TAMs from a larger patient cohort, as well as through analysis of pre- and on-treatment biopsies from patients who received MOv18 IgE in the Phase I trial. All of these analyses recapitulate the pro-inflammatory activating effects of IgE on patient macrophages and underline its clinical relevance.

Point 3: In humans, there are multiple myeloid subsets that express FcεRI. Why is every dotplot specifically gated on CD14+ cells? FcεRI is expressed, for instance, in human dendritic cells (as suggested by genomic data), and of course in mast cells. Expression of FcεRI in these immune cells in cancer patients, plus different types of true TAMs, should be investigated by flow cytometry, and not merely by CIBERSORT; much less relying on TCGA datasets, which restricted ovarian cancer analyses to those with >70% tumor cell nuclei enrichment. The analysis of in vitro (artificially) generated macrophages must be complemented with real world samples, which are much more heterogeneous and complex.

Authors' Response

Our study focuses on evaluating the effects of the clinically studied MOv18 IgE on macrophages from patients with epithelial ovarian cancers. This work builds upon extensive pre-clinical evidence of the anti-tumoural efficacy for IgE mAbs, via macrophage-mediated Fc functions, in several published studies (Karagiannis et al., *J Immunol*, 2007, 179(5):2832-43; Josephs et al., *Cancer Res*, 2017, 77(5):1127-1141 & *Allergy*, 2018, 73(12):2328-2341; Pellizzari et al., *EBioMedicine*, 2019, 43:67-81; Chauhan et al, *Nat Commun*, 2023, 14(1):2192).

In concordance, our scRNA-seq analysis of immune cells from metastatic peritoneal tumours, included in the original manuscript, demonstrated monocytes and macrophages to be the most abundant myeloid cell type and to exhibit highest *FCER1A* expression (please see Figure 2e and text at lines 239-242). The same analyses, pointed to low levels of dendritic cell (DC) types present, which expressed low *FCER1A*, whilst no mast cells were identified. In our revised manuscript:

- We characterise these known FcεRI-expressing cell types in ovarian cancer patient ascites by flow cytometry (please see new Figure 2g, bottom, and new text at lines 249-252). In accordance with scRNA-seq findings: a) TAMs were the most frequent cell type; b) low levels of DCs were identified and; c) no mast cells were detected.
- We also demonstrate consistent TAM expression of FcεRI (please see new Figure 2g, top and new text at lines 246-247 and 524-531) and CD23 (please see new Supplementary Figure 2d and new text at lines 246-247) at the protein level, again in line with the scRNA-seq identification of a *FCER1A*+ TAM subset across patients (Figure 2f and text at lines 242-245).
- To further address the Reviewer's point, we also further evaluated IgE-mediated activation of additional FcεRI-expressing immune cells, which may contribute to MOv18 IgE effector function:
 - We demonstrated that MOv18 IgE binds human DCs; and MOv18 IgE, but not a non-specific IgE isotype control, promotes DC phagocytosis of FRα-coated beads (please see new Supplementary Figure 5a-b and new text at lines 337-341 and 575-579); these findings indicate the potential for MOv18 IgE to promote DC antigen presentation; consistent with data presented previously in an extensive study focusing on DC-IgE interactions (Platzer B et al., *Cell Rep*, 2015, 10(9):1487-1495).
 - Moreover, we identified very low levels of basophils in patient ascites and demonstrated that basophils undergo IgE-mediated activation by upregulation of the cell surface marker CD63 (please see new Supplementary Figure 5c-d and new text at lines 341-344).

In summary, in our revised manuscript we have now completed a broad characterisation of IgE engagement and activation of TAMs, and additional myeloid cells, in ovarian cancer, using high-dimensional flow cytometry and scRNA-seq, rather than relying on CIBERSORT cell type imputation. Together, our findings support a principal role for TAMs in IgE effector functions,

as the central theme of our present study, whilst demonstrating the potential for a contribution from additional myeloid cell types.

We would like to reiterate that the purpose of this study was not to characterise the engagement and activation of the whole TME myeloid cell compartment by MOv18 IgE. Across disparate *in vivo* models, our group has identified MOv18 IgE, and tumour-specific IgE mAbs (e.g., Chauhan et al., *Nat Commun*, 2023, 14(1):2192), to exhibit a principal monocyte and macrophage-mediated anti-tumoural mechanism. With respect to MOv18 IgE, there are multiple lines of evidence that point to a central role for monocytes and macrophages in its anti-tumoural functions: 1) In an orthotopic patient-derived xenograft (PDX) nude mouse model engrafted with human PBMCs, MOv18 IgE-treated PDXs displayed increased intratumoural human macrophages, without enrichment for additional human immune cell types (*Figure 4a*) (Karagiannis et al., *Eur J Immunol*, 2003, 33(4):1030-40); 2) In this PDX mouse model, the survival advantage provided by MOv18 IgE compared to controls, was abrogated when monocytes were depleted from the engrafted PBMCs (*Figure 4b*) (Karagiannis et al., *J Immunol*, 2007, 179(5):2832-43); 3) In a syngeneic lung metastasis rat model, which we also transcriptomically evaluated in this manuscript (*Figure 8a-c*), rats treated with MOv18 IgE displayed enhanced metastases reduction compared to control, which was associated with increased intratumoural macrophage infiltration and reduced FR α -expressing cancer cells (*Figure 4c-d*) (Josephs et al., *Cancer Res*, 2017, 77(5):1127-1141). We include examples of these data for the Review below.

Our finding in this revised manuscript from on-treatment biopsies from the Phase I trial of MOv18 IgE, of increased intratumoural CD68 expression, is consistent with and further supports the significance of macrophages in IgE-mediated anti-tumoural functions (please see *Figure 8d* and new text at lines 454-458 and 617-623).

[figure redacted]

Figure 4: Monocytes and macrophages have been identified as central effector cells in MOv18 IgE studies.

(a) A patient-derived xenograft (PDX) mouse model, comprising orthotopic intraperitoneal xenografts produced by challenge with ovarian cancer patient ascites and an infusion of human PBMCs, was treated with MOv18 IgE and PBS control. Immunohistochemical examination of macrophage infiltration in paraffin-embedded PDX sections. Top: Densities of human CD68+ macrophages (cells/mm³). Bottom: Representative immunohistochemistry images. Human macrophages were rarely observed in PDX from mice treated with PBMC alone (a), compared with those treated with PBMC and MOv18 IgE (b) (taken from Karagiannis et al., *Eur J Immunol*, 2003, Figure 2, 33(4):1030-40). (b) Effect of monocytes and MOv18 IgE on survival of mice of the same PDX model (taken from Karagiannis et al., *J Immunol*, 2007, Figure 6, 179(5):2832-43). (c) An immunocompetent lung metastases rat model, produced through systemic injection of the syngeneic WAG rat adenocarcinoma cell line CC531 transfected with human FR α , treated with mouse/rat chimeric MOv18 IgE, mouse/rat chimeric MOv18 IgG or PBS control. Paraffin-embedded lung sections show increased CD68+ macrophages and reduced FR α + tumour cells in tumour islets, following MOv18 IgE treatment compared to MOv18 IgG and PBS control. (d) MOv18 IgE treatment is associated with superior lung metastasis reduction compared to MOv18 IgG and PBS control (taken from Josephs et al., *Cancer Res*, 2017, Figure 2c, 3c, 77(5):1127-1141).

Point 4: Binding of IgE to other cell types different from TAMs or DCs outside the TME is also relevant to understand potential toxicities in future trials. PBMCs from ovarian cancer patients should be also analyzed, if the goal is to determine the global effects of this Ab.

Authors' Response

We thank the Reviewer for raising this point. The purpose of our study was to characterise the phenotype of ovarian cancer patient TAMs and assess how this is modulated by IgE-mediated activation, especially in the patient TME. Characterisation of the systemic effects of IgE mAbs in circulating immune cells including monocytes, PBMCs, basophils and eosinophils has been extensively studied by our group in several pre-clinical models, including rodent tumour models and human blood (Karagiannis et al., *J Immunol*, 2007, 179(5):2832-43; Josephs et al., *Cancer Res*, 2017, 77(5):1127-1141 & *Allergy*, 2018, 73(12):2328-2341; Nakamura et al., *Cancers*, 2020, 12(11):3376; Bax et al., *Br J Cancer*, 2023, 128(2):342-353; Chauhan et al., *Nat Commun*, 2023, 14(1):2192). We have also addressed the safety of IgE mAbs in several studies in the blood of both healthy volunteers and patients of different solid tumours, including ovarian cancer (Rudman et al., *Clin Exp Allergy*, 2011, 41(10):1400-13; Bax et al., *Allergy*, 2020, 75(8):2069-2073). Ultimately, we have reported on the safety of MOv18 IgE in the Phase I clinical trial of MOv18 IgE (NCT02546921) (Spicer et al., *Nat Commun*, 2023, 14(1):4180). As discussed in the previous responses to the Reviewer's points, instead we sought to characterise its principal anti-tumoural mechanism, via macrophage-mediated Fc effector function.

With respect to the characterisation of the IgE-mediated engagement and activation of cancer patient PBMCs, we have previously demonstrated Fc ϵ RI expression and IgE mAb binding in monocytes from patients with ovarian cancer (Nakamura et al., *Cancers*, 2020, 12(11):3376). Additionally, we have found IgE-engaged PBMCs and monocytes, from cancer patients of different solid tumour types, to exhibit ADCC-mediated killing of cancer cells (Josephs et al., *Cancer Res*, 2017; Nakamura et al., *Cancers*, 2020, 12(11):3376; Chauhan et al., *Nat Commun*, 2023, 12(11):3376). We demonstrated that monocytes undergo an IgE-mediated pro-inflammatory activation to produce a cytokine release signature associated with improved survival in ovarian cancer patients (Nakamura et al., *Cancers*, 2020, 12(11):3376). In our revised manuscript, we additionally characterised IgE-mediated engagement and activation of human DCs, which can be localised both in the circulation and TME. As discussed, we demonstrated that MOv18 IgE binds DCs and induces DC phagocytosis of FR α -coated beads

in an antigen-specific manner (please see new Supplementary Figure 5a-b and new text at lines 337-341 and 575-579). These findings are consistent with a previous in-depth study of DC-IgE interactions (Platzer et al., *Cell Rep*, 2015, 10(9):1487-1495).

Point 5: Interactions between T cells and macrophages are allogeneic, which could change subtle effects in an autologous setting.

Authors' Response

We thank the Reviewer for raising this important point about the co-culture assays. In our revised manuscript we now include autologous co-cultures between macrophages and naïve CD4⁺ T cells. In accordance with our findings from allogeneic co-cultures, we identified macrophages to promote Treg induction from naïve CD4⁺ T cells, with this effect reversed by FcεR:IgE cross-linking with polyclonal anti-IgE antibodies. Consistent with allogeneic co-cultures, Treg induction was restricted when macrophages were FcεR:IgE cross-linked for 1 hour prior to co-cultures, and was abrogated when cross-linking was maintained for the 3-day co-culture (please see new Supplementary Figure 6c and new text at lines 402-403).

Point 6: Do hyperinflammatory TAMs induced by MOv18 IgE have the capacity to cross-present antigen (e.g., DQ-OVA)? Or merely alter cytokine production and the expression of co-stimulatory molecules?

Authors' Response

We thank the Reviewer for this interesting question. In our revised manuscript, we demonstrated IgE-mediated pro-inflammatory repolarisation of patient ascites TAMs. This pro-inflammatory repolarisation included an upregulation of T cell co-stimulatory molecules CD40 and CD80 (and a trend towards HLA-DR upregulation) (please see updated Figure 4g and updated text at lines 318-324), as well as an upregulation of a hyperinflammatory subset enriched for CD40, CD80, CD86 and HLA-DR (please see updated Figure 5e-g and updated text at lines 364-369 and 573-575).

The IgE-mediated upregulation of T cell co-stimulatory molecules and the subset-level enrichment of HLA-DR may indicate the potential for enhanced TAM antigen presentation following FcεR:IgE cross-linking. Macrophage:T cell co-cultures, including the autologous co-cultures included in the revised manuscript, were conducted with a CD3/CD28 agonist (TransAct, Miltenyi), a T cell activation effect independent of MHC:T cell receptor (TCR)-mediated activation (Figure 6-7 and text at lines 377-430). This allowed the evaluation of how the overall repolarisation of the phenotype of patient-derived macrophages by IgE, affects their immunosuppressive interactions with Tregs, which we identified to be enriched in patient tumours (Figure 6a-c, Supplementary Figure 6 and text at lines 381-391). This demonstrated that IgE-mediated repolarisation restricted Treg induction by patient-derived macrophages, which subsequently inhibited Treg suppression of CD8⁺ T cells downstream (Figure 7 and text at lines 407-430).

As IgE-mediated repolarisation of patient-derived macrophages promotes supportive conditions for CD8⁺ T cell proliferation, the capacity for MOv18 IgE to induce macrophage cross-presentation to CD8⁺ T cells is an area of interest. In the revised manuscript, we have provided early investigation of this in cells known to cross-present antigen, DCs. We demonstrated that MOv18 IgE, but not a non-specific isotype control, induced DC phagocytosis of FRα-coated beads (please see new Supplementary Figure 5a-b and new text at lines 337-341 and 575-579). Internalisation of antigens is the first requisite step in antigen presentation and this may indicate its promotion. Our findings included in the revised manuscript are consistent with a previous study which extensively evaluated DC-IgE interactions and demonstrated: ovalbumin (OVA):anti-OVA IgE internalisation to cross-presentation endosome compartments in human DCs; and activation of OVA-specific CD8⁺ T cells in mice following injection of DCs cross-linked by OVA:anti-OVA IgE *ex vivo* (Platzer et al., *Cell Rep*, 2015, 10(9):1487-1495). However, unlike DCs, the ability of macrophages to

cross-present antigen is not established in humans. In fact, TAMs from OVA⁺ melanoma mouse tumours were shown to engage OVA-specific CD8⁺ T cells via an antigen-specific MHC:TCR synapse, but this was found to trigger T cell exhaustion rather than proliferation (Kersten et al., *Cancer Cell*, 2023, 26;40(6):624–638.e9).

In our study we are investigating macrophages in the context of MOv18 IgE:FR α in humans, and therefore the use of an OVA-based experimental system is unsuitable. Instead, this will require the identification of specific patients with FR α -specific CD8⁺ T cell responses, followed by co-cultures between MOv18 IgE:FR α cross-linked patient macrophages and sorted autologous FR α -specific CD8⁺ T cells. This will require extensive development, including a patient screening tool via FR α peptide libraries which can bind MHC class I, and subsequently, fluorescently-tagged MHC class I:peptide tetramers (HLA matched) to sort FR α -specific CD8⁺ T cells before co-culture and cross-presentation experiments.

Consequently, this project is beyond the scope of the current study, however it will be important to investigate it in future studies. Nonetheless, in our revised manuscript we have demonstrated: a) low levels of DCs in ovarian cancer patient ascites (please see new Figure 2g, bottom, and new text at lines 250-252); b) human DC binding to MOv18 IgE; and c) MOv18 IgE-specific activation of human DCs (please see new Supplementary Figure 5a-b and new text at lines 337-341). Our findings, consistent with a previous study of DC-IgE interactions (Platzer et al., *Cell Rep*, 2015, 10(9):1487-1495), provides initial indications of the potential for this axis to contribute to the macrophage-mediated anti-tumoural mechanism of MOv18 IgE, which we have included in our revised Discussion at text lines 573-579.

Point 7: Figure 8 is of doubtful relevance, because the pattern of expression of Fc ϵ RI in humans is different from rodents. In addition, MOv18 IgE should induce neutralizing rat Abs, which could complicate the interpretation of the results.

Authors' Response

The development and findings of this IgE-treated syngeneic lung metastases rat model, from which the transcriptomic data in Figure 8 was generated from, were published in Josephs et al., *Cancer Res*, 2017, 77(5):1127-1141 & *Allergy*, 2018, 73(12):2328-2341.

Data of treatment with rat MOv18 IgE in this rat model were accepted by MHRA, the UK Regulator, as the regulatory authority-approved model for the formal MOv18 IgE toxicology (conducted by an MHRA-appointed Contract Research Organisation, CRO) and efficacy studies. These allowed the initiation of the MOv18 IgE Phase I clinical trial in patients with solid tumours (most of whom were patients with ovarian cancer).

The syngeneic immunocompetent WAG rat model of cancer entails: a) challenge with a syngeneic WAG rat tumour and b) systemic treatment with the surrogate rat MOv18 IgE.

We would like to clarify the following:

- Fc ϵ RI expression is substantially different between humans and mice.
 - Tetrameric Fc ϵ RI ($\alpha\beta\gamma_2$) is expressed in both mice and humans by mast cells and basophils.
 - Mice lack expression of trimeric Fc ϵ RI ($\alpha\gamma_2$).
 - Humans exhibit trimeric Fc ϵ RI expression in eosinophils, DCs and monocytes and macrophages; the expression of which is absent in these cells in mice.
- Fc ϵ RI expression is comparable in humans and rats:
 - Like in humans, rats exhibit both the trimeric Fc ϵ RI ($\alpha\gamma_2$) and the tetrameric Fc ϵ RI ($\alpha\beta\gamma_2$) forms of Fc ϵ RI (Dombrowicz D et al., *J Immunol*, 2000, 165(3):1266-71).
 - Therefore, the rat immune system, as demonstrated by our group and others (Dombrowicz D et al., *J Immunol*, 2000, 165(3):1266-71; Josephs et al., *Cancer Res*, 2017, 77(5):1127-1141 & *Allergy*, 2018, 73(12):2328-2341) closely recapitulates the pattern of human Fc ϵ RI expression.

- Rat macrophages express a functional FcεRI, bind rat IgE and perform IgE-mediated protective immune functions observed in humans, including ADCC-mediated killing of parasites via IgE:FcεRI (Dombrowicz D et al., *J Immunol*, 2000, 165(3):1266-71).
- In this rat tumour model, we re-engineered MOv18 IgE (a mouse/human chimeric antibody) to bear a rat constant region, producing a mouse/rat chimeric MOv18 IgE (rMOv18 IgE). We found no evidence of anti-drug antibodies (ADAs) responses, even at doses of up to 50 mg/kg MOv18 IgE (10 rats receiving 4 weekly doses of rMOv18 IgE) (*Allergy*, 2018, 73(12):2328-2341).
- Consistent with this, we demonstrated a lack of ADA responses in the MOv18 IgE clinical trial (Spicer et al., *Nat Commun*, 2023, 14(1):4180).

The rat tumour model is therefore highly suitable for the study of both efficacy and safety for IgE and mirrored findings in the Phase I clinical trial of MOv18 IgE.

We have updated the text to clarify the close recapitulation of human FcεRI expression in the rat immune system and the use of a mouse/rat chimeric MOv18 IgE in this model, at lines 441-442 and 440, respectively.

Minor Comments:

Point 1: In the Introduction, the authors made a statement about how the affinity of FcεRI for IgE should increase the half-life of IgEs, compared to IgGs. That is not factually correct, because, for instance, the half-life of Nivolumab (an IgG4 that binds very weakly to Fc Receptors) is >20 days, which appears to depend on FcRn.

Authors' Response

We thank the Reviewer for highlighting the need to clarify this point. We and others have demonstrated IgE antibodies to exhibit a short half-life in the serum compared to IgG antibodies (<1.5 days compared to 14-20 days for IgG) (Lawrence et al., *J Allergy Clin Immunol*, 2017, 139(2):422-428.e4; Spicer et al., *Nat Commun*, 2023, 14(1):4180). The statement in the original manuscript of an increased half-life for IgEs compared to IgGs was in reference to the tissue half-life rather than serum half-life. For example, IgE has a reported half-life in the skin of approximately 14 days, compared to approximately 3 days for IgG (Lawrence et al., *J Allergy Clin Immunol*, 2017, 139(2):422-428.e4). This distinction was however not clear in the original manuscript and we have now clarified this in our revised manuscript (please see the updated text at lines 116-117).

Reviewer's Comments
Point-by-point Responses

Reviewer #1 Comments:

General Comment: The authors have conducted extensive additional analyses in response to the provided comments. Most notably, they have increased the patient sample size to strengthen the findings. They have also demonstrated that TAMs exhibit an immunosuppressive profile similar to MAsc, which is modulated by IgE to induce a hyperinflammatory phenotype. Additional results from clinical trial patients, showing an IgE-immune activation signature, enhance the translational relevance. The revised manuscript includes improved flow cytometry data presentation.

Authors' Response

We thank the Reviewer very much for their positive review of our manuscript.

Point 1: While the manuscript has been significantly improved, some concerns remain, primarily related to the modest sample sizes in certain assays and inconsistencies in cohort sizes for some parameters.

Authors' Response

We thank the Reviewer for raising this point.

With respect to modest sample sizes: the transcriptomic data from the rat model of lung metastases (Figure 8a-c) were produced for our previously published study by Josephs et al., *Allergy*, 2018, 73(12):2328-2341 and were analysed for the purpose of this manuscript. In our revised manuscript, we included immunohistochemical analysis of pre- and on-treatment biopsies of metastatic tumours from 2 patients enrolled in the first-in-class Phase I clinical trial of MOv18 IgE (NCT02546921) (Figure 8d). These samples were provided through our patients kindly volunteering to undergo invasive procedures. Data from these analyses were in accordance with transcriptomic data from IgE-treated rats: we identified evidence of a recapitulation of an IgE-immune activation signature in patients during MOv18 IgE treatment, through increased intratumoural localisation of immune cells and CD68 and CD3 expression in both patients. The hypothesis of the syngeneic rat tumour model analysis was to evaluate whether the IgE-immune activation signature, identified in *ex vivo* IgE stimulation assays of patient-derived macrophages and T cells, was recapitulated in an immunocompetent *in vivo* cancer context. Therefore, we believe that the Phase I data compliments the animal data, and significantly expands the translational relevance of it and the overall study; through direct assessment of the effect of MOv18 IgE on cancer patients. We also undertook the challenge of sourcing more ascites samples from ovarian cancer patients and consequently increased the cohort size in all of the assays on ascites-isolated TAMs. This included the functional characterisation of the effect of MOv18 IgE on TAMs, via *ex vivo* FcεR:IgE cross-linking with polyclonal anti-IgE, in which we increased the number of patients examined from n=7 to n=13 (Figure 4g). This confirmation in a larger cohort size of an IgE-mediated pro-inflammatory repolarisation of ascites TAMs away from their immunosuppressive phenotype, increased the translational relevance of our findings for therapeutic IgE antibodies.

With respect to differences in cohort sizes: the evaluation of IgE-mediated activation of *in vitro*-derived macrophage subsets in Figure 1e, utilised different flow cytometry panels, resulting in different n numbers depending on the marker. Scavenger receptors were stained for in all donors (n=9), whereas FcεRI (n=6) and T cell cross-talk molecules (n=5) were not stained for in all donors. However, for the characterisation of ascites-conditioned macrophages (MAsc) and ascites TAMs, a consistent flow cytometry panel was used for each cell type, resulting in the same n number for markers in each experiment.

In our manuscript, we have clarified the n numbers for all analyses in the text and figure legends. Furthermore, we include a Source Data file containing the data points for each figure.

Reviewer #3 Comments:

General Comment: The main concern of reviewer 2 was related to the artificially induced macrophages and their relevance in MOv18. In this new version authors show 1) expression of Fc γ R in TAM 2) MOv18 binding TAM 3) TAM repolarization to inflammatory anti-tumor profile by MOv18 binding. In my opinion, these three findings are sufficient to prove MOv18 activity on TAMs and relevance of this therapy in ovarian cancer associated macrophage polarization.

Authors' Response

We thank the Reviewer very much for their positive review of our manuscript.

Point 1: The authors could also provide additional evidence from the literature demonstrating the conserved phenotype and function of ascites macrophages and true TAMs?

Authors' Response

We thank the Reviewer for raising this point. In our revised manuscript, we have included a discussion of published studies which describe a conservation of immunosuppressive TAM phenotypes between ascites and primary and metastatic tumours, and their association with IL-10. Please see new text lines 490-494. This is consistent with our analyses of ascites (ascites-conditioned macrophages (MAsc) and ascites TAMs) and primary (TCGA-OV; bulk RNA-seq) and metastatic tumours (GSE165897; single cell RNA-seq).

Point 2: The nomenclature used to describe these macrophage populations is a bit inconsistent throughout the manuscript. Calling them simply patient macrophages or ovarian cancer patient macrophages is somewhat misleading and the alternatively used identifiers in vitro-derived, ascites-conditioned or ascites TAMs are better for clearly differentiating the source of the macrophages.

Authors' Response

We thank the Reviewer for highlighting this point. In our revised manuscript, we have removed the term 'patient macrophages' when referring to ascites-conditioned macrophages (MAsc) and ascites TAMs collectively. Instead, we have either referred to them specifically or when collective description is necessitated, we have used the term 'patient-derived' for improved clarity; this term is clearly defined in the updated text lines 123-125.

Point 3: Additionally, the (n) of samples for ascites TAMs is different across several assays so the authors should confirm the consistency of sample numbers.

Authors' Response

Thank you for raising the need for this clarification. As the ascites TAMs assays were run with fresh and not cryopreserved ascites specimens, it was not possible to use TAMs from each patient in every assay. This was additionally limited by an insufficient cell number being obtained from certain ascites specimens, due to biological variability. Please see Supplementary Table 2 for a description of the patients used in each ascites TAMs assay.

Point 4: Authors included autologous co-cultures, however, is not correctly confirmed in method section that both CD4 and macrophages are isolated from the same individual. Please include that description.

Authors' Response

We thank the Reviewer for highlighting the need for clarification here. Please see updated text lines 775-776.

Reviewer #4 Comments:

General Comment: This manuscript by Osborn et al., examines the repolarisation of macrophage subsets from several sources by IgE and induction of a hyperinflammatory phenotype that restricts macrophage:Treg interactions. The monoclonal IgE antibody mOv18 (targeting folate receptor-alpha) is currently being examined in a first-in-class clinical trial. This study attempts to understand the mechanisms of IgE-mediated macrophage modulation, extending on preclinical rodent studies.

The study is highly relevant to the understanding of a IgE monoclonal antibodies for cancer immunotherapy. If proven effective in early phase trials, it will be vital to understand the immunological mechanisms unique to using IgE mAbs compared to more standard approaches and how these can be taken advantage of therapeutically. Therefore, this should be a significant piece of work of interest to experts in the field on immuno-oncology.

The revised conclusions and additional work conducted, particularly on the ascites TAMs, help to validate the conclusions and findings of the manuscript.

Authors' Response

We thank the Reviewer very much for their positive review of our manuscript.

Point 1: In their revised analysis the authors could place more emphasis on the ascites derived TAM studies and less of the MAsc experiments and provide additional evidence from the literature linking ascites TAM phenotype with that of intra-tumoural TAMs, as ascites samples are often used for biomarker analysis, one would assume a significant correlation between the two sources.

Authors' Response

We thank the Reviewer for raising this point. In our revised manuscript, we have included a discussion of published studies which describe a conservation of immunosuppressive TAM phenotypes between ascites and primary and metastatic tumours, and their association with IL-10. Please see new text lines 490-494. This is consistent with our analyses of ascites (ascites-conditioned macrophages (MAsc) and ascites TAMs) and primary (TCGA-OV; bulk RNA-seq) and metastatic tumours (GSE165897; single cell RNA-seq).

Point 2: The authors should take care to clarify the source of macrophages used throughout and keep consistent nomenclature.

Authors' Response

We thank the Reviewer for highlighting this point. In the revised manuscript, we have referred to ascites-conditioned macrophages (MAsc) and ascites TAMs specifically or when collective description is necessitated, they are denoted by 'patient-derived' rather than 'patient' macrophages; with 'patient-derived' clearly defined at new text lines 123-125.

Comments to editors and review of responses to reviewer #2 comments - NCOMMS-23-60207A

In response to the editors request I have reviewed the manuscript #NCOMMS-23-60207A. In particular, I have assessed the comments of the previous reviewer #2 and the revisions completed by the authors in answering the reviewers' questions. I have addressed some of the reviewers' concerns, but should state, I do not know the recommendation of their review. I have also included my own brief evaluation of the immunology and sequencing data presented in the revised manuscript.

In response to Reviewer #2's major comments:

The reviewer's major issue is the use of ex-vivo polarised macrophages (MAsc) rather than true TAMs (e.g. obtained from patient biopsy samples), which may not completely recapitulate the complexity and microenvironment in which these cells arise. They note that obtaining these samples and repeating experiments may not be feasible within scope of the study/trial. They also request analysis of other FcεRI+ cell types not in the initial analysis, which the authors address in the revisions.

In general, I agree with reviewer, particularly that response of macrophages to IgE in culture may be very different that within the tumour itself where other cell types and signals are at play. However, if biopsy samples were not available in the study design the question is: do the authors do enough to prove that MAsc are a valid model of tumour-derived TAMs. This is supported by literature and perhaps the groups previous publications and helped by the addition of macrophages derived from ascites fluid as a comparator, as these are likely to be a better representative of true TAMs. I think given the conclusions drawn from these experiments are not overreaching that these experiments are fine, given the relevant caveats are acknowledged and supporting literature for this technique is cited. The authors could also provide additional evidence from the literature demonstrating the conserved phenotype and function of ascites macrophages and true TAMs?

The nomenclature used to describe these macrophage populations is a bit inconsistent throughout the manuscript. Calling them simply patient macrophages or ovarian cancer patient macrophages is somewhat misleading and the alternatively used identifiers in vitro-derived, ascites-conditioned or ascites TAMs are better for clearly differentiating the source of the macrophages. Additionally, the (n) of samples for ascites TAMs is different across several assays so the authors should confirm the consistency of sample numbers.

1. Expression of target FOLR1 on other cells in the TME – The reviewer asks for additional information to rule out FOLR1 expression on immune cells within ovarian cancers. The authors provide reasonable data from sc-RNA-seq datasets showing FOLR1 expression is restricted to tumour cells. They follow this with flow cytometry data of TAMs and tumour cells present in ascites fluid demonstrating a clear expression of FR-alpha on tumour cells but not TAMs. While this does not rule out the presence, or induction, of folate receptor on TAMS within the tumour, it provides evidence for a clear differentiation between the targeted tumour cells and TAM effector cells.

2. In vitro-derived macrophage populations – The reviewer points out that the artificially-induced macrophages may not be a suitable representative of TAMS present

within tumour given their unique ontogeny and microenvironment, which cannot be recapitulated *ex vivo*. The authors have completed extensive work to attempt to prove similarity in phenotype between their *in vitro*-derived and ascites-conditioned macrophages and validated TAM populations, with new data providing comparison to ascites TAMs. The authors should be careful with the use of the wording 'TAM' even when reporting on TAMs from ascites fluid, without clear data linking the phenotype and function of these cells to TAMs found within ovarian cancers. However, the data support a clear phenotypic similarity between these subsets and conservation in response. Whether the response to treatment *ex vivo* will be mirrored in TAMs within tumours remains to be seen.

3. Multiple cell types express FcεRI e.g. dendritic cells – The reviewer points out that other subsets of cells express FcεRI within the complex tumour microenvironment and requests additional validation. The authors again turn to ascites samples as a window to the TME and examine FcεRI expression on the cells contained in the ascites, showing low levels of expression on some DC subsets. I am happy with the additional analyses performed in answering the reviewer's request.

4. Binding of IgE to cells outside the TME is relevant for toxicity studies – The reviewer suggests that analysing IgE binding across cell types in human PBMCs to assess global effects and toxicity of the antibody. The authors point out that studies examining the binding of IgE mAbs to circulating immune cells are already present in the literature and point to several references, including their own previous work. I believe this should satisfy the reviewer, given it is a broad request and not really the focus of the current manuscript.

5. T cell – macrophage interactions are allogeneic – The reviewer points out an issue with the macrophage: T cell co-culture system which is addressed in the revisions by the authors.

6. Do activated TAMs have the ability to cross-present antigen – The reviewer asks for additional functional data on the hyperinflammatory TAMs. The authors respond by detailing the cytokine and receptor phenotype of these hyperinflammatory TAMs and detail why they feel they cannot conduct the appropriate experiment (lack of experimental system/reagent) to test the reviewer's query. While the question is interesting, it is questionable if the results from a contrived *in vitro* assay would fully replicate the reality *in vivo*. While potentially of interest this is not the focus of the manuscript and does not need to be completed for publication.

7. Differing patterns of FcεRI in humans and rodents. The reviewer states that the expression differences for FcεRI between humans and rodents invalidate the data in Figure 8. The authors refute this assertion, citing references to show, while the mouse immune system differs, the rat immune system is more similar to that of humans. They also clarify the use of a mouse/rat chimeric Mov18 IgE that does not result in anti-drug antibody formation. I am satisfied this response justifies the models used and data evaluated.

For my own evaluation of the manuscript, focused mainly on immunology and flow/seq analyses; I found the data to be well presented, the figures clear and analyses well planned

out. It was particularly nice to see representative FACS plots showing antibody staining and immune subsets alongside summary statistics, which is now often overlooked in many publications. Therefore, I have no reason to doubt the veracity to the data and the study looks to have been conducted to a high standard.

The authors are forced somewhat to pull data from several sources across transcriptomics databases, in vitro assays and ex vivo analyses leaving some gaps in direct correlation, but this is no longer unusual for such studies. Sample sizes appear sufficient for statistical analyses. Additionally, as the therapy is already advanced to the clinical phase (although the current data presented (Fig 8) is quite limited) and has reasonable preclinical data supporting safety and efficacy the approach seems immunologically sound. There is some valid criticism, pointing out the limitations of the in vitro models employed, and the relation to actual phenotype and function of TAMs within the TME, but it is unclear whether the authors had access to a better alternative. I do not have any serious concerns with the study design or data analysis, as the manuscript has already been through one round of peer review.